# Jigsaw: Learning to Assemble Multiple Fractured Objects

**Jiaxin Lu** [*]     **Yifan Sun** [*]     **Qixing Huang**
Department of Computer Science
University of Texas at Austin
{lujiaxin, yifansun12}@utexas.edu     huangqx@cs.utexas.edu

## Abstract

Automated assembly of 3D fractures is essential in orthopedics, archaeology, and our daily life. This paper presents Jigsaw, a novel framework for assembling physically broken 3D objects from multiple pieces. Our approach leverages hierarchical features of global and local geometry to match and align the fracture surfaces. Our framework consists of four components: (1) front-end point feature extractor with attention layers, (2) surface segmentation to separate fracture and original parts, (3) multi-parts matching to find correspondences among fracture surface points, and (4) robust global alignment to recover the global poses of the pieces. We show how to jointly learn segmentation and matching and seamlessly integrate feature matching and rigidity constraints. We evaluate Jigsaw on the Breaking Bad dataset and achieve superior performance compared to state-of-the-art methods. Our method also generalizes well to diverse fracture modes, objects, and unseen instances. To the best of our knowledge, this is the first learning-based method designed specifically for 3D fracture assembly over multiple pieces. Our code is available at https://jiaxin-lu.github.io/Jigsaw/.

## 1 Introduction

The task of assembling 3D fractures has extensive applications across numerous fields. For instance, orthopedic doctors need to realign dislocated bone fragments, and subsequently create bone plates and screws to heal compound fractures. Archaeologists, on the other hand, need to recreate the original shape and functionality of unearthed artifacts by assembling the fractures. These procedures demand significant expertise, are prone to errors, and can be tedious. Even in the context of daily life, furniture assembly can be a challenging and exhausting task that requires an understanding of mechanical structures and component matching. In the past two decades, many efforts attempted to address the challenge of automatic assembly. Traditional methods apply hand-crafted geometric features to detect the fracture surfaces and optimize pairwise matching among these surfaces [1, 2]. Recently the availability of large scale 3D datasets [3, 4, 5, 6] have boosted learning based frameworks for solving 3D assembly tasks. Semantic-aware methods [7, 8, 9, 10, 11, 12] target at assembly from semantically segmented parts and predict semantic labels as matching priors. Geometry based methods [13, 14, 15] leverage fracture shapes and continuity of textures in the procedure of piece matching.

However, in the context of fracture assembly for restoring broken objects, there is no guarantee that individual pieces will retain semantic meanings. Texture information may also be either non-accessible or lacking, such as on glass bottles or worn artifacts. This calls for a general 3D assembly solver that utilizes hierarchical features of both the global piece surfaces and local geometry shapes of fractures. In this work, we introduce a **Jo**int Learning of **S**egmentation and **A**lignment Frame**w**ork

---

[*]Equal Contribution

37th Conference on Neural Information Processing Systems (NeurIPS 2023).

(**Jigsaw**), for assembling objects damaged or shattered due to physical impact or force. Given a set of fractured pieces represented as point clouds without texture, our method recovers the global pose of each piece to restore the underlying object.

Our Jigsaw framework consists of four parts: (1) A front-end feature extractor with self-attention and cross-attention layers for local geometric feature extraction. (2) Categorize the surface of each fractured piece into two segments: the indiscernible fracture surface and the visible original surface. (3) A novel formulation for multi-piece assembly that learns a bipartite matching among points on the fracture surface from all pieces. (4) Recover pairwise pose based on the learned correspondences and perform robust global alignment to compute the global poses of all pieces. Key features of Jigsaw are that 1) it learns two correlated tasks, i.e., segmentation and matching among all involving pieces, jointly, and 2) it seamlessly integrates feature matching and the rigidity constraint among consistent features, We evaluate the effectiveness of our framework on Breaking Bad [4]. Experimental results demonstrate the effectiveness of our method in surface segmentation, fracture point matching, and significantly outperforming state-of-the-art methods[8, 16]. The main contribution includes the following:

- We propose Jigsaw, a novel joint learning framework tailored for multi-part fracture assembly. Our approach embodies an attention-based feature extractor network to accurately capture local geometry features of each point. Additionally, we introduce a primal-dual descriptor that effectively captures viewpoint-dependent characteristics for surface matching.
- Jigsaw incorporates fracture point segmentation to capture intrinsic features, employs a novel multi-part matching formulation to establish automatic piece positioning within one object, and utilizes global alignment for accurate global pose alignment.
- Experimental evaluation on the multi-part assembly Breaking Bad dataset demonstrates the superior performance of Jigsaw compared to baseline models, showcasing its strong generalization ability to unseen objects. We further highlight the limitations of baseline models that rely on global features, which we argue are too abstract for this task and lack generalizability.
- To the best of our knowledge, Jigsaw is the first learning-based method specifically designed for the assembly of multiple pieces from physically broken 3D objects.

## 2   Related Works

**Feature matching**. Early works apply hand-crafted features over fracture surfaces for fracture matching between different pieces [17, 1, 2, 18]. These features are also used to identify the fractured part from the entire surface [17, 1, 2]. However, hand-crafted features are in lack of robustness for assembly tasks over large datasets due to different materials and fracture patterns of objects. In recent years, deep learning methods have gained significant traction in matching problems. Methods utilizing CNN, GNN, and attentions have found successful applications in various domains, such as image registration [19, 20] and graph or multi-graph matching on images [21, 22, 23, 24, 25]. While their efficacy in these scenarios is noteworthy, the majority of these methods predominantly focus on simpler settings and have not adequately addressed the challenges in multi-part 3D fracture assembly.

**Part assembly**. Semantic-aware learning methods have been highly successful in the task of part assembly. [9, 11] are designed for assembling specific CAD mechanics. For categorical everyday objects, [26, 10, 27, 12, 28] generate the missing parts based on the accumulated shape prior to completing the entire object, which can result in shape distortion from the input parts. [8, 7] apply graph learning to predict part labels and assembly orders. All of these methods require the input objects decomposed in a semantically consistent way and need specific training for each category of objects. Fracture assembly poses unique challenges due to the variety of objects and the lack of semantic meanings associated with individual pieces, adding to the complexity of the task.

**Geometry based learning methods**. Recent approaches have aimed to capture geometry information using deep features for piece matching. [13, 15] combine local geometry with textures for feature generation, but may suffer when texture information is not accessible, as in most point cloud representations. [14] apply a transformer for local shape encoding and an adversarial network to help generate a plausible assembly of two pieces. However, for multiple fracture assembly, objects can break into 5 pieces or more, with the largest piece smaller than half of the original object. Under this setting, training an adversarial network to evaluate the quality of assembly becomes less effective.

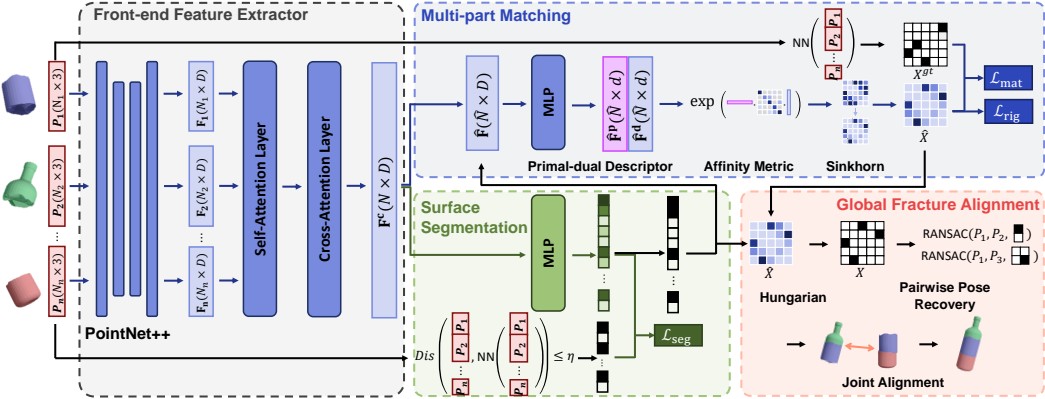

Figure 1: Overall pipeline for Jigsaw (mesh used only for visualization). The method consists of four parts: front-end feature extractor, surface segmentation, multi-part matching, and global fracture alignment. In the front-end feature extraction, we use a multi-scale grouping PointNet++ [31], and one self-attention layer followed by one cross-attention layer to extract features for each point. Surface segmentation is used to locate all fracture points in one broken object, and multi-part matching finds correspondences of fracture points among multiple pieces. The matching results will be used for pairwise pose recovery and joint alignment to retrieve an assembled object. More detail about each part will be discussed in Section 3.

**Low-overlap 3D registration**. Another relevant research field is low-overlap 3D registration. Recent works have shown the potential of data-driven methods in registration tasks with approximately 30% overlap [29, 16, 30]. [16, 30] learn a classifier to determine the overlapped sections and predict inter-piece point matching inside the sections, while [29] using object semantic to guide registration. The fracture assembly task can be viewed as an even more extreme case of registration, where the overlap between pieces can fall below 4%.

## 3   Joint learning framework for 3D fracture assembly

Given a set of fractured pieces $\mathcal{P} = \{P_1, P_2, \cdots, P_n\}$ represented as point clouds uniformly distributed on the surface, our goal is to recover the 6-DoF pose $\{T_1, T_2, \cdots T_n\}$ in $SE(3)$ for each piece and restore the underlying object $O = T_1(P_1) \cup T_2(P_2) \cup \cdots \cup T_n(P_n)$, where $T_i(\cdot), 1 \le i \le n$ is the operator to recover $P_i$ to its original position by transformation $T_i$. In the fracture assembly setting, the object $O$ is a rigid body, and the pieces are the result of physical cracking or breakage without any deformation. Also, no pieces are lost, which ensures that the restored pieces can approximately reconstruct the entire object $O$.

To handle this challenge, we propose Jigsaw, a learning-based framework that jointly optimizes surface segmentation and fracture matching among all pieces of the object. The entire assembly network consists of a segmentation module and a matching module that shares a front-end feature extractor. The segmentation module uses only the intrinsic shape information of each piece to separate the fracture surface from the original surface. The matching module exploits the primal dual descriptor of the affinity metric to propagate mutual information among all pieces and establish the matching between fracture points (that is, points on the fracture surface) of different pieces via Sinkhorn [32]. With the fracture point matching predicted by the network, we recover the pairwise transformations and perform global alignment with standard approaches. The complete pipeline of our framework is shown in Fig. 1. Note that this design ensures that segmentation and matching are performed jointly. In the subsequent part of this section, we will first introduce the front-end extractor, then delve into the details of the segmentation module, followed by the multi-part matching module and post-processing for pose recovery.

### 3.1   Front-end Feature Extractor

In contrast to previous methods [33, 4] that utilize global piece-wise descriptors for assembly, we focus on geometric features. As illustrated in Figure 7 of the PointNet paper [34], a global descriptor remains the same as long as the piece lies between the learned critical point set and the upper-bound

point set. This indicates that global descriptors are coarse and insufficient to represent the intricate geometry of the fracture surface, which is crucial for our task. To address this limitation, we employ a multi-scale grouping PointNet++ [31] as the backbone of our feature extractor to capture local geometry features, denoted as $\mathbf{f}_p \in \mathbb{R}^D$ from each point $p \in P_i$.

Furthermore, We emphasize the significance of relative positional information between points and pieces for identifying the non-smooth fracture surfaces, enforcing rigidity during matching, and accurately placing the pieces. To facilitate the integration of such intra-piece and inter-piece information, we leverage transformer layers. Specifically, we introduce both self-attention and cross-attention layers as tools to reason about the relative information between points.

For the self-attention within a piece, we employ a single point transformer layer [35]. The self-attention mechanism is defined as follows:

$$\mathbf{f}_p^s = \sum_{\mathbf{f}_q \in \mathcal{N}(p)} \mathtt{softmax}(\mathtt{MLP\_s}((W_Q^s \mathbf{f}_p - W_K^s \mathbf{f}_q + \mathbf{p}^{\mathrm{enc}}))) \odot (W_V^s \mathbf{f}_q + \mathbf{p}^{\mathrm{enc}}) \tag{1}$$

Here, $W_Q^s, W_K^s, W_V^s \in \mathbb{R}^{D \times D}$ are weights for query, key, value in the attention layer, $\mathbf{p}^{\mathrm{enc}}$ represents the positional encoding for point $p_i$, obtained using a Multi-Layer Perceptron (MLP). Unlike the standard dot-product attention layer, it employs vector weights: $\mathtt{MLP\_s}$ is a mapping function that produces the weight vector, $\mathtt{softmax}(\cdot)$ normalizes the weights, and $\odot$ denotes element-wise multiplication. $\mathcal{N}(p) \in P_i$ represents the neighborhood of point $p$, calculated using $k$-nearest neighbors for local feature aggregation.

For cross-attention, we use standard multi-head attention with position-wise feed forward over the entire object to facilitate the communication of local features among different pieces. Let $\mathbf{F}^s \in \mathbb{R}^{N \times D}$ pack all the point features of one object after the self-attention, and $\mathbf{F}^c \in \mathbb{R}^{N \times D}$ denote the point features produced by cross-attention,

$$\mathbf{F}^c = \mathrm{FFN}(W_O^m(\mathrm{cat}(\mathrm{head}_1, \ldots, \mathrm{head}_h))), \text{ where } \mathrm{head}_i = \mathrm{Attention}(W_{Q\ i}^m \mathbf{F}^s, W_{K\ i}^m \mathbf{F}^s, W_{Q\ i}^m \mathbf{F}^s)$$

$$\text{and } \mathrm{Attention}(Q, K, V) = \mathtt{softmax}(\frac{QK^T}{\sqrt{d_k}})V \tag{2}$$

Here, $W_{Q\ i}^m, W_{K\ i}^m, W_{V\ i}^m \in \mathbb{R}^{D \times d_h}, W_O^m \in \mathbb{R}^{hd_h \times D}$ denote the weights for projection. The function $\mathrm{FFN}(\cdot)$ consists of two linear functions with ReLU activation in between, followed by a LayerNorm as normalization.

## 3.2 Surface segmentation

The fracture assembly problem can be viewed as a special case of the 3D registration problem, where the overlapping between adjacent pieces only lies in the fracture surface between them. Let $P_i^f \subset P_i$ be the subset of points on fracture surfaces of $P_i$ and let $P_{ij} \subset P_i^f$ denote the subset of points on the fracture surface between $P_i$ and $P_j$ for arbitrary two pieces $P_i, P_j$. For pieces $P_i$ and $P_j$ adjacent in their original pose $T_i(P_i)$ and $T_j(p_j)$, we have $P_{ij} \neq \emptyset, P_{ji} \neq \emptyset$. Under the assumption that the point clouds are uniformly distributed, we can approximate the relative pose $T_{ij}^\star = T_j^{-1} T_i$ between $P_i, P_j$ with

$$T_{ij}^\star = \operatorname*{argmin}_{T_{ij} \in SE(3)} d\left(P_{ji}, T_{ij}(P_{ij})\right), \tag{3}$$

where $d(\cdot, \cdot)$ is some distance function between two point clouds. (3) becomes the standard formulation for point cloud registration with perfect prior for the overlapping. Before we can optimize (3) directly, we need to figure out the overlapping part $P_{ij}$ and $P_{ji}$ for each pair $(P_i, P_j)$. Although $P_{ij}$ is dependent on both pieces, previous studies [17, 1, 2] have demonstrated that the segmentation of fracture surfaces $P_i^f$ of $P_i$ is an intrinsic property that can be inferred from the local shapes of each $P_i$ individually. However, these methods require a continuous smooth surface to compute hand-crafted shape features for accurate classification. To capture local geometric properties under the discrete point cloud setting where surface normal and curvature are no more available, we introduce our deep surface segmentation module to handle the surface segmentation task.

The surface segmentation module can be viewed as a binary classifier of each point. Let $c_p$ be the indicator function that determines whether a point $p$ of a point cloud $P \in \mathcal{P}$ is a fracture point. For each 3D model uniformly sampled as $N = \sum_{i=1}^{n} N_i$ points, the surface segmentation module

takes the position of the points $P_i \in \mathbb{R}^{N_i \times 3}$ of a piece $P_i \in \mathcal{P}$ and predicts a confidence score $c_i$ to segment a point $p \in P_i$ into the fracture surface.

Our front-end feature extractor introduced in Section 3.1 has extracted local geometry feature $\mathbf{f}_p \in \mathbb{R}^D$ for each point. Next, we use two MLP layers to reduce the number of channels to 1, followed by a sigmoid function to predict the confidence $c_p$ for labeling $p$ as a fracture point. We set the segmentation loss $\mathcal{L}_{\text{seg}}$ to be a negative log-likelihood loss, which is supervised by the ground truth label $c_p$ for every point $p$ as

$$\mathcal{L}_{\text{seg}} = -\frac{1}{N}\sum_{p \in O} c_p \log \tilde{c}_p + (1-c_p)\log(1-\tilde{c}_p). \tag{4}$$

The ground truth label $c_p$ for each point $p \in P_i$ is constructed by examining its distance to its nearest neighbor in other parts:

$$c_p = \begin{cases} 1 & Dis(p, \mathsf{NN}(p, \mathcal{P}\backslash P_i)) \leq \eta, \\ 0 & \text{otherwise.} \end{cases} \tag{5}$$

where $Dis(\cdot)$ is a distance function and $\mathsf{NN}(\cdot, \cdot)$ is to find the nearest neighbour of one point in a point set.

Although the fraction of the fracture surface area may differ between objects, and the positive and negative samples in (4) may be unbalanced, we have found that adding two MLP layers after feature encoding help to yield good performance for the segmentation task. The segmentation module is able to accurately predict fracture points even in cases with a large number of pieces or very small fracture surfaces, and we leave the evaluation details of this module in the appendix A.

## 3.3 Multi-part Matching

One key aspect of Jigsaw is its ability to combine multiple pieces. Traditional approaches to assembling pieces rely on pairwise matching, which involves assessing the compatibility of two parts, identifying corresponding points within them, and aligning those points. However, this method is prone to cumulative errors, and a single mistake can ruin the entire assembly task. In addition, matching small pieces with limited geometry information solely based on pairwise information is extremely challenging. Consequently, the pairwise information in the multipart fracture assembly dataset may not be comprehensive enough to capture the true matching possibilities between all the pieces involved in the assembly process. Even advanced low-overlap 3D registration methods like PREDATOR [16], which share a similar structure, have failed to handle assembly tasks effectively, as shown in Section 4.2. Thus, alternative approaches are needed to address the limitations of pairwise matching, and we introduce our multipart matching module.

Facilitated by the global fracture points of a segmented object in Section 3.2, we present a global view to match all its pieces simultaneously. We observe that each fracture surface should have one precise match within a broken object, which is a rule that applies specifically to the fracture piece assembly task. Therefore, we learn the matching among all pieces without specifying their pairwise relationship. This allows each fracture point to automatically find a similar counterpart on a different piece solely based on local geometry. Moreover, the correspondences highlight the assembly regions, avoiding the need for costly predictions of fine-grained pairwise overlapping areas.

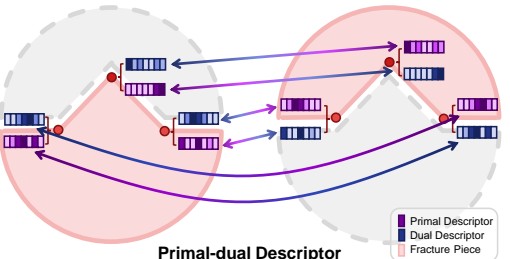

Figure 2: Illustration of Primal-dual descriptor: The red region and points represent the target fragment piece and the points, respectively. The purple feature (primal descriptor) on the left learns the local convex shape of the piece, while the blue feature (dual descriptor) on the right captures the concavity observed from the outside. The primary and dual descriptors on both sides will be matched in the matching module.

**Primal-dual Descriptor.** Intuitively, when an object has been broken into two pieces by physical force, the two pieces should exhibit complementary geometry. Conventional feature representation is used to find similar geometry and, therefore, would match two identical surfaces, which becomes undesirable in the context of fracture assembly. With this concern, we propose the primal-dual descriptor, designed to capture the essence of complementary geometry (see Fig. 2).

Let $\{\hat{\mathbf{f}}_i^c\}_{i=1}^{n_c}$ be the selected features of the fracture points extracted from the backbone network. These features encode the local geometry of the fracture surfaces. We apply MLP layers and a normalization layer to $\hat{\mathbf{f}}_i^c$ and get the primal descriptor $\hat{\mathbf{f}}_i^p$ and dual descriptor $\hat{\mathbf{f}}_i^d$:

$$\hat{\mathbf{f}}_i^p, \hat{\mathbf{f}}_i^d = \text{norm}(\text{MLP\_aff}(\hat{\mathbf{f}}_i^c)). \tag{6}$$

The primal-dual descriptor learns surface features that capture the characteristics of a local surface from both directions. It promotes the robustness and reliability of surface matching and avoids learning biased features based on a single viewing direction. We leave more detailed analysis to the appendix B.

**Affinity Metric.** Let $\hat{\mathbf{F}}^p, \hat{\mathbf{F}}^d \in \mathbb{R}^{\hat{N} \times d}$ collect all the fracture points features extracted from descriptors, we compute the global affinity matrix $M \in \mathbb{R}^{\hat{N} \times \hat{N}}$ and the respective doubly-stochastic matrix $\tilde{X} \in [0,1]^{\hat{N} \times \hat{N}}$ as follows:

$$M = \exp\left(\frac{\hat{\mathbf{F}}^{p\top} A \hat{\mathbf{F}}^d}{\tau}\right), \quad \tilde{X} = \text{Sinkhorn}(M), \tag{7}$$

Here, $A \in \mathbb{R}^{d \times d}$ consists of learnable affinity weights, and $\tau$ denotes the temperature parameter [36]. We then apply the Sinkhorn layer [32], which is a differentiable operation, to obtain a doubly stochastic matrix $\tilde{X} \in [0,1]^{\hat{N} \times \hat{N}}$. This matrix represents the soft matching predicted by the multipart matching module.

**Matching Loss.** Given the ground truth position of each part, we construct the ground truth matching matrix $X^{gt} \in \{0,1\}^{\hat{N} \times \hat{N}}$. Here, $x_{ij}^{gt} = 1$ if and only if the $j$-th fracture point is the nearest neighbor of the $i$-th fracture point and both points belong to different pieces; otherwise, $x_{ij}^{gt} = 0$. To compute the matching loss, we employ the cross entropy between $\tilde{X}$ and $X^{gt}$:

$$\mathcal{L}_{\text{mat}} = -\frac{1}{\hat{N}} \sum_{1 \leq ij \leq \hat{N}} x_{ij}^{gt} \log \tilde{x}_{ij} + (1 - x_{ij}^{gt}) \log(1 - \tilde{x}_{ij}) \tag{8}$$

With the ground truth matrix $X^{gt}$ generated based on finding the nearest neighbor of each point, the matching loss will also enforce a global rigidity guidance that each node should be matched to its neighbor.

**Rigidity Loss.** As 3D fracture assembly is applied to rigid objects, we further enforce the rigidity loss over the pairs of matched pieces. Let $\tilde{X}_{ij} \in \mathbb{R}^{\hat{N}_i \times \hat{N}_j}$ be the submatrix of $\hat{X}$ that indicates the likelihood between the fracture points on the piece $P_i$ and $P_j$. For a fracture point $p$ in $P_i$, $\tilde{X}_{ij}$ matches it to a point $p'$ computed as the weighted average of all the matchable points in $P_j$:

$$p' = \frac{\tilde{X}_{ij}(p) P_j}{\|\tilde{X}_{ij}(p)\|_1}, \tag{9}$$

where $\tilde{X}_{ij}(p) \in \mathbb{R}^{1 \times \hat{N}_j}$ is the row in $\tilde{X}_{ij}$ that contains matching likelihood from $p$ to $P_j$. We optimize the transformation $\tilde{T}_{ij}$ from $P_i$ to $P_j$ by minimizing the weighted mean squared error between the matched fracture points

$$\tilde{R}_{ij}, \tilde{\boldsymbol{t}}_{ij} = \underset{R, \boldsymbol{t}}{\arg\min} \sum_{p \in P_i} \|\tilde{X}_{ij}(p)\|_1 \|R(p) + \boldsymbol{t} - p'\|_2. \tag{10}$$

where $\tilde{R}_{ij}, \tilde{\boldsymbol{t}}_{ij}$ are the rotation and translation part of $\tilde{T}_{ij}$ and $R(\cdot)$ is the rotation operator. (10) has a closed-form solution:

$$(U, \Sigma, V^\top) = \text{SVD}\left(\sum_{p \in P_i} \|\tilde{X}_{ij}(p)\|_1 p^\top p'\right), \ \tilde{R}_{ij} = UV^\top, \ \tilde{\boldsymbol{t}}_{ij} = -\frac{\sum_{p \in P_i} \|\tilde{X}_{ij}(p)\|_1 (R(p) - p')}{\sum_{p \in P_i} \|\tilde{X}_{ij}(p)\|_1}. \tag{11}$$

The rigidity loss is computed as:

$$\mathcal{L}_{\text{rig}} = \sum_{1 \leq i,j \leq n} \mathcal{R}_{ij}, \text{ where } \mathcal{R}_{ij} = \sum_{p \in P_i} \|\tilde{X}_{ij}(p)\|_1 \|\tilde{R}_{ij}(p) + \tilde{\boldsymbol{t}}_{ij} - p'\|_2. \tag{12}$$

To avoid numerical instability and improve efficiency in (11) during training, we compute $\tilde{T}_{ij}$ in each iteration without involving back-propagation and only set $\tilde{X}_{ij}$ as the learnable variable.

The loss for training is composed as:

$$\mathcal{L} = \alpha\mathcal{L}_{\text{seg}} + \beta\mathcal{L}_{\text{mat}} + \gamma\mathcal{L}_{\text{rig}}. \tag{13}$$

**Multi-part Matching** With the doubly-stochastic matrix $\tilde{X}$, computing multi-part matching results becomes straightforward. Given the fact that the primal and dual feature represents the surface looked from a different viewpoint, they should be distinct and not assigned good confidence for matching. Therefore, we can directly pass $\tilde{X}$ to a Hungarian layer [37]. This yields a binary permutation matrix $X$, representing a bipartite matching among all fracture points:

$$X = \texttt{Hungarian}(\tilde{X}) \tag{14}$$

As mentioned, the resulting matching matrix $X$, along with the affinity matrix $M$ and soft matching matrix $\tilde{X}$, captures the local geometry similarity across different pieces. Additionally, they quantify the confidence level of how well the pieces fit together. This information enables an accurate positioning and alignment of the fractured components.

### 3.4 Global Fracture Alignment

To efficiently and robustly recover the global poses of each fractured piece of object $O$, we adopt a two-step pipeline based on the bipartite matching $X \in \{0, 1\}^{\hat{N} \times \hat{N}}$ predicted by the multipart matching module.

In the first step, we compute pairwise transformations between each pair of pieces $(P_i, P_j)$ to remove outlier matches. Let $X_{ij} \in \mathbb{R}^{\hat{N}_i \times \hat{N}_j}$ denote the submatrix of $X$, which establishes the correspondences between the fracture points from $P_i$ and $P_j$. We apply the RANSAC algorithm [38] to compute the transformation $\tilde{T}_{ij}$ from $\hat{P}_i$, $\hat{P}_j$ and $X_{ij}$. This step ensures reliable pairwise transformations for subsequent alignment.

In the second step, we perform robust global alignment using the computed pairwise transformations. We model the global alignment configuration as a factor graph [39], denoted as $G = (\mathcal{V}, \mathcal{E})$, where the global poses $\tilde{T}_i$ are optimized on the vertices $\mathcal{V}$ and the pairwise transformations $\tilde{T}_{ij}$ are set as constraints on the edges $\mathcal{E}$. The information matrix $I(e)$ over the edge $e = (i, j)$ is set to $|X_{ij}|_{\text{Fro}}^{-2} \cdot I_6$, with $I_6$ being the $6 \times 6$ identity matrix. We employ Shonan averaging [40], a state-of-the-art 6-DoF global alignment method, to optimize global poses over $G$. The global alignment method takes pairwise transformations between potentially adjacent pieces as input and outputs a pose for each piece up to a global transformation over the coordinate system. To avoid such an ambiguity, we anchor the coordinate system to the canonical one of the largest pieces in our experiment and measure the errors over all other pieces of the fractured object. Further technical details of this module can be found in the references [41, 42, 43]. Please refer to those sources for more information, as it is beyond the scope of our main contribution.

## 4 Experiments

We demonstrate the effectiveness of our 3D fracture assembly framework through experimental evaluations on a large-scale fracture assembly dataset. Our results show that Jigsaw significantly outperforms the baseline methods both quantitatively and qualitatively. Additionally, we conduct ablation studies to analyze the contributions of each module in our framework. All experiments are conducted on a Linux workstation with 4 Tesla V100-SXM2-32GB GPUs, Intel(R) Xeon(R) CPU E5-2698 v4 @ 2.20GHz CPUs, and 480GB Memory. Table 1 lists the configurations of the parameters in our experiments. A comparison of the training/testing time is included in Appendix D.

### 4.1 Protocols

**Dataset.** We leverage the Breaking Bad dataset (Sellan et al., 2022), a novel data set of multiple fracture assemblies featuring synthetic physical breaking patterns. Our training was on `everyday` subset and the testing was on both `everyday` and `artifact` subsets for a fair comparison with

Table 1: The detailed experiment parameters. We follow the parameters provided in [16, 33, 4] to reproduce their results. Training, model, and dataset parameters are included.

| | Jigsaw | Predator[16] | DGL [33] | LSTM [33] | Global [33] | description |
|---|---|---|---|---|---|---|
| epoch | 250 | 200 | 200 | 200 | 200 | training epochs |
| bs | 4 | 16 | 32 | 32 | 32 | batch size |
| lr | 0.001 | 0.01 | 0.001 | 0.001 | 0.001 | learning rate |
| optimizer | Adam | SGD | Adam | Adam | Adam | optimizer during training |
| scheduler | Cosine | Exponential | Cosine | Cosine | Cosine | learning rate scheduler |
| min_lr | 1e-5 | - | 1e-5 | 1e-5 | 1e-5 | minimum learning rate for Cosine scheduler |
| $\alpha$ | 1.0 | - | - | - | - | segmentation loss ratio |
| $\beta$ / e | 1.0 / 10 | - | - | - | - | matching loss ratio change at epoch e |
| $\gamma$ / e | 1.0 / 200 | - | - | - | - | rigidity loss ratio change at epoch e |
| $\tau$ | 0.05 | - | - | - | - | temperature paramter for affinity |
| sampling by | object | piece | piece | piece | piece | sampling strategies for point clouds |
| points ($N$) | 5000/o | 800/p | 1000/p | 1000/p | 1000/p | points sampled per object (/o) or piece (/p) |
| $\eta$ | 0.025 | 0.025 | - | - | - | segmentation ground truth label threshold |

baselines. `Everyday` consists of 498 models and 41,754 fracture patterns, and is split into a training set (34,075 fracture patterns from 407 objects) and a test set (7,679 fracture patterns from 91 objects). `Artifact` consists of 3651 fracture patterns from 40 uncategorized objects. The average diameter of the objects in the training and testing dataset was 0.8. Categorical information was concealed during all the experiments.

The input was uniformly sampled $N$ points on the surface of the object as a point cloud. In composing the point cloud, two sampling strategies are considered, sampling by piece and object. Sampling by piece, as used in the Breaking Bad benchmark [4], equally samples points in each fragment, which would cause point density imbalance. To better reflect real-world scanning, we opt for "sampling by object", sampling a fixed number of points in each object, while sampling points for each fragment based on its surface area. We ensure at least 30 points per fragment for multi-part matching. Additional processing details can be found in the Appendix C.1.

**Baseline Approaches.** As learning to assemble multiple fractured objects is a novel task, there is a dearth of existing methods for direct comparison. Therefore, we have selected a set of established methods that primarily address similar challenges in 3D alignment and assembly, and employ them as baselines for comparative evaluation. Global [26, 10] extracts per-piece features, which are combined with global shape descriptors to regress the pose of each piece in one shot. LSTM applies a bidirectional LSTM similar to [27] and estimates the pose of each piece in sequential style. DGL [8] is a state-of-the-art approach for the assembly task of parts and can also be adapted to the assembly task of fractures. It leverages an iterative graph neural network to reason about the relationships among pieces. Additionally, we take PREDATOR [16] as a minor competitor, which is a state-of-the-art approach for 3D registration with low overlapping rates. It employs a multitask transformer to estimate the overlap between pairs of pieces in conjunction with their relative pose. All baseline approaches use textureless point clouds as input and are trained under the `everyday` object subset of the Breaking Bad dataset.

**Evaluation Metrics.** We adopt the same evaluation schemes for 3D assembly tasks used in [4, 14, 10]. We report the mean absolute error (MAE) and the root mean square error (RMSE) for the rotations and translations of the estimated global poses. Additionally, we include the part accuracy (PA) metric proposed in [33], which measures the ratio of perfectly assembled pieces based on an average Chamfer distance of less than 0.01 for each point between the assembly results and the ground truth.

## 4.2 Performance

**Overall Performance.** We report the performance of Jigsaw and all the baseline methods over the 3D fracture assembly task on both `everyday` and `artifact` object subsets. An overall quantitative comparison of the evaluation metrics is presented in Table 2. For PREDATOR we apply the evaluation metrics on pairwise transformations since it only supports overlapped pairs of pieces as input.

Jigsaw significantly outperformed all the baselines across all the evaluation metrics. On `everyday`, we achieved an average rotation error of 36.3° over all pieces, which was a 46% reduction compared to the top-performing baseline method, DGL. Our transformation error was also outstanding, with an average of $8.7 \times 10^{-2}$, surpassing the best baseline method with an error of $11.8 \times 10^{-3}$. We

Table 2: Quantitative results of baseline methods and Jigsaw on the Breaking Bad dataset.

| Method | Original Task | RMSE (R) ↓ degree | MAE (R) ↓ degree | RMSE(T) ↓ ×10⁻² | MAE (T) ↓ ×10⁻² | PA ↑ % |
|---|---|---|---|---|---|---|
| Results on the `everyday` object subset. | | | | | | |
| Global [26, 10] | assembly | 82.4 | 69.7 | 14.8 | 11.8 | 21.8 |
| LSTM [27] | assembly | 84.7 | 72.7 | 16.2 | 12.7 | 19.4 |
| DGL [8] | assembly | 80.6 | 67.8 | 15.8 | 12.5 | 23.9 |
| DGL(5000/o) [8] | assembly | 81.1 | 68.1 | 15.4 | 12.3 | 25.5 |
| PREDATOR [16] | registration | 82.8 | 71.2 | 12.5 | 10.2 | 1.3 |
| Jigsaw (Ours) | assembly | **42.3** | **36.3** | **10.7** | **8.7** | **57.3** |
| Results on the `artifact` object subset. | | | | | | |
| Global [26, 10] | assembly | 86.9 | 75.3 | 17.5 | 14.5 | 5.6 |
| LSTM [27] | assembly | 85.6 | 74.1 | 18.6 | 15.2 | 4.5 |
| DGL [8] | assembly | 86.3 | 74.3 | 18.0 | 14.9 | 9.6 |
| PREDATOR [16] | registration | 86.0 | 74.8 | **13.4** | **10.9** | 1.1 |
| Jigsaw (Ours) | assembly | **52.4** | **45.4** | 22.2 | 19.3 | **45.6** |

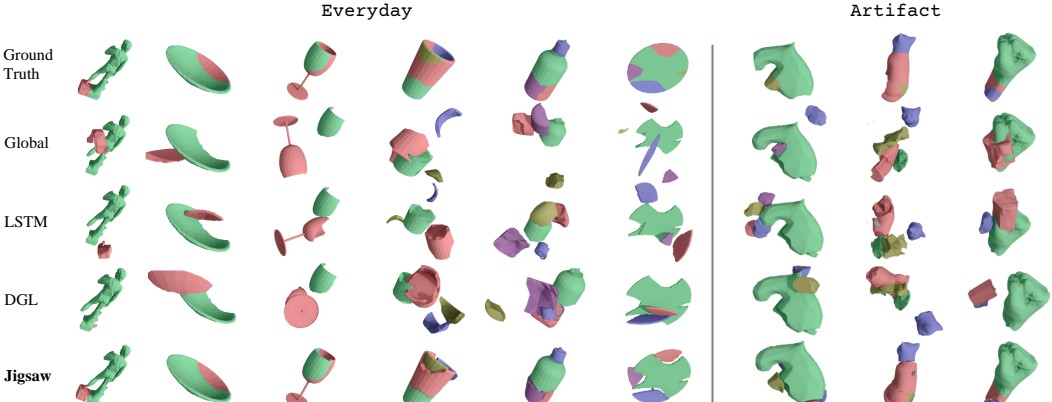

Figure 3: Qualitative results of baseline methods and Jigsaw on the Breaking Bad dataset (mesh used only for visualization). The coordinate system of the green piece is set as the global coordinate system. Better viewing with color and zooming in.

successfully restored over $57\%$ of the pieces to their original pose while baseline methods restored only half of us. We also show that change in sampling strategy has had no impact on the outcome of baseline methods through a sampling by object version of DGL. Our method exhibited remarkable generalizability on `artifact`, showing a slight increase of $10.1°$ in average rotation error and achieving a part accuracy of $44\%$. In contrast, all baseline methods completely failed to handle new categories of models that were not present in the training dataset. The PREDATOR model failed to accurately identify the pairwise overlap region, resulting in erroneous predictions of complete overlap between the two pieces. While this may seem to yield a small translation error, it failed to perform adequately in the assembly scenario. Hence, we exclude it from further discussion.

**Detailed distribution of each metric.** Fig. 4 presents a comprehensive analysis of each metric for all assembly models. Regarding rotation and translation metrics, a larger surface area under the curve indicates better performance. Jigsaw exhibits a significantly larger number of samples with small rotation and translation errors, as illustrated in the figure. Although Jigsaw reports a slightly larger translation error than the baseline models on the `artifact` object subset, the distribution results reveal that this is primarily due to a few extreme cases, leading to an undesirable accumulation of the translation error. In terms of part accuracy, a larger surface area above the curve signifies superior results. Notably, Jigsaw demonstrates a higher proportion of correctly positioned pieces.

**Visualization.** The qualitative comparison of different models is shown in Fig. 3. We plot the recovered object from different models. It demonstrates that our model could find the position of the fracture pieces and recover a good pose of each piece. Visualization of objects from different categories also implies our model has good generalization ability and it doesn't require object-level information as assembly guidance.

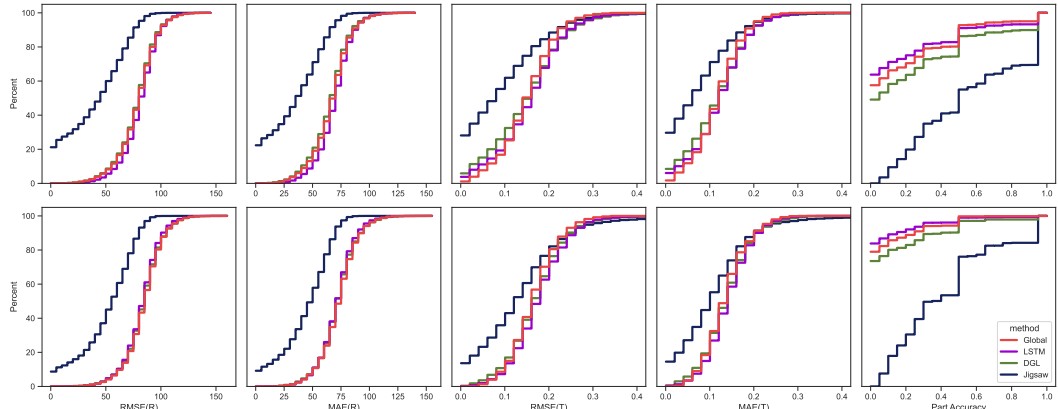

Figure 4: The discretized distributions of metrics evaluated over the test data of `everyday` (up) and `artifact` (bottom) object subset in the Breaking Bad dataset.

**Ablation Study.** We conducted an ablation study to assess the effectiveness of each component in our Jigsaw model. The baseline model consists of a segmentation module and a multi-part matching module, trained separately using separate backbones and without incorporating transformers. The plain joint training model was trained using the same loss function as the Jigsaw model, but without using two transformer layers. The effectiveness of the rigidity loss regularization was evaluated over the complete network with both joint training and transformer layers.

As presented in Table 3, the joint training model outperformed the baseline in transformation metrics. This is due to a more favorable initialization for the matching module achieved through pretraining the backbone with the segmentation module. Joint training also aided the matching module's convergence to better optimum. Attention layers significantly reduced rotation errors:

Table 3: Ablation study of Jigsaw.

| Components | | | RMSE (R) | MAE (R) | RMSE (T) | MAE (T) | PA |
|---|---|---|---|---|---|---|---|
| Joint Training | Attention | Rigidity | degree | degree | $\times 10^{-2}$ | $\times 10^{-2}$ | % |
| | | | 51.9 | 44.6 | 32.4 | 27.2 | 45.7 |
| ✓ | | | 51.7 | 44.4 | 12.7 | 10.2 | 47.0 |
| ✓ | ✓ | | 42.3 | 36.3 | 10.9 | 8.9 | 57.2 |
| ✓ | ✓ | ✓ | **42.3** | **36.3** | **10.7** | **8.7** | **57.3** |

the self-attention layer embedded geometric information into point features, while the cross-attention layer improved multi-part matching by incorporating information from other parts. Applying rigidity loss as a refinement additionally brought an improvement of $2 \times 10^{-3}$ in translation. Even the baseline method outperformed previous works, highlighting the effectiveness of our segmentation and multi-part matching modules.

## 5 Conclusion and Limitations

In this study, we propose a novel geometry-aware framework to address the 3D fracture assembly task. Our approach involves a joint learning model that effectively extracts local geometry features from the backbone and enables the simultaneous learning of piecewise fracture surface segmentation and global fracture point matching. Experimental evaluations and thorough analyses substantiate the exceptional performance of our method in terms of accurately recovering piece poses and restoring the original object. Furthermore, we demonstrate the generalizability of our approach across objects of diverse categories. One limitation of our approach lies in that as the number of fractured pieces increases, the accuracy of pose estimation tends to decrease due to potential ambiguities in fracture matching caused by local geometric features. Future research directions may include exploring matching pruning techniques for pairwise pose estimation to prevent penetration between fractured pieces and enforcing smoothness constraints to enhance pose accuracy.

## Acknowledgements

We would like to acknowledge NSF IIS-2047677, HDR-1934932, and CCF-2019844.

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

# Appendix

## A  Details of Network Architecture

We provide additional information about our network architecture. The backbone of our network is based on a standard PyTorch implementation of multi-scale grouping PointNet++. It is followed by a single MLP layer that extracts a $D$-dimensional feature for each point. In our implementation, we set $D = 128$.

The detailed structure of the transformer module can be seen in Fig. 5. The self-attention layer follows the official code of the point transformer layer, while the cross-attention layer employs a multi-head attention mechanism with position-wise feed-forward networks. In our configuration, we set the number of attention heads to $h = 8$, the head dimension to $d_h = 16$, and use $k$-nearest neighbor sampling with $k = 16$. The inner layer of the feed-forward network has a dimensionality of $d_i = 256$.

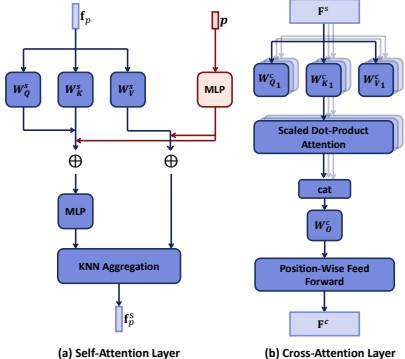

Figure 5: Detailed structure of the attention layers.

Regarding the primal-dual descriptor, we set its feature dimension to $d = 256$.

## B  Analysis on Primal-dual Descriptor

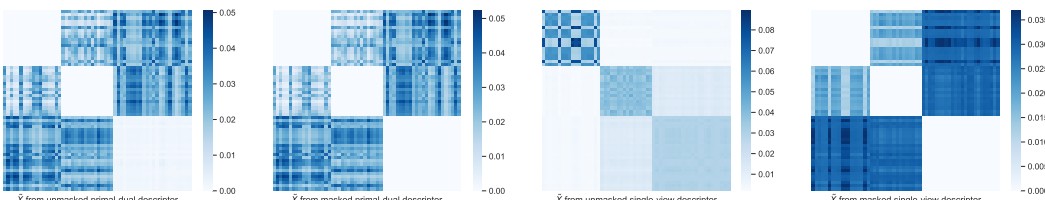

Figure 6: Comparison between the doubly-stochastic soft matching matrix $\tilde{X}$ computed by different types of descriptors.

We conducted an analysis to compare the results obtained using the primal-dual descriptor and the standard single-view descriptor. Before passing the affinity metric computed from the single-view descriptor to the Sinkhorn layer, a common practice is to mask out the diagonal sub-matrix to prevent self-self alignment. For clarity, we refer to the masked and unmasked versions of the primal-dual and single-view descriptors to denote whether this masking operation has been applied or not.

In Fig. 6, we provide a visualization of the doubly-stochastic matrix $\tilde{X}$ computed by the Sinkhorn algorithm using four types of descriptors: (1) the unmasked primal-dual descriptor, (2) the masked primal-dual descriptor, (3) the unmasked single-view descriptor, and (4) the masked single-view descriptor. The visualization depicts an example object with 3 pieces and 58 fracture points. The $\tilde{X}$ obtained from the primal-dual descriptor clearly demonstrates its ability to differentiate between the two viewpoints and avoid aligning a point to itself. In contrast, the single-view descriptor exhibits a diagonal peak, resulting in self-self alignment. Even when the diagonal sub-matrix is masked to prevent self-self alignment, the soft matching computed from the single-view descriptor is less distinct compared to that computed from the primal-dual descriptor.

Furthermore, we conducted evaluation on the `everyday` object subset of Breaking Bad dataset, using the unmasked primal-dual descriptor and the masked primal-dual descriptor. The results indicate that the unmasked primal-dual descriptor achieves comparable performance to the masked one, with only a slight difference: a decrease of $0.2°$ in rotation error (MAE(R)) and an improvement of $0.7 \times 10^{-2}$ in translation error (MAE(T)). We attribute this minor difference to the fine characteristics of the primal-dual descriptor that prevent self-self alignment, thus resulting in consistent performance.

# C Experiment Details

## C.1 Dataset

We leverage Breaking Bad dataset [4] to evaluate our method and all baseline methods. As we have stated in section 4, the training of all methods were on the training subset of `everyday`, and testing were on the testing subset of both `everyday` and `artifact` object. Each object within the dataset has been fragmented into pieces, represented by triangle meshes, and all the pieces are in their original poses. By assembling these pieces together directly, the surface of the original object is seamlessly restored. The triangle meshes of the pieces solely consist of exterior faces that are visible from the outside.

In generating the point cloud for our experiments, we employ two sampling strategies for the compared methods: "sampling by piece" and "sampling by object". The "sampling by piece" strategy, originally utilized in the Breaking Bad benchmark [4], involves sampling an equal number of points within each fragment. However, this approach leads to excessively dense sampling on small fragments, while larger fragments suffer from sparser point distributions, resulting in an imbalance in the representation of point density across fragments. To better mirror real-world scanning technology, we opt for the "sampling by object" strategy. With this approach, we sample a fixed number of points within each object, and the number of sampled points for each fragment is determined based on its surface area. In essence, smaller fragments receive fewer points, whereas larger ones receive more, ensuring a more realistic representation. Additionally, we ensure that each fragment is sampled with a minimum of 30 points to include even the tiniest fragments in multi-part matching. Detailed parameter settings can be found in Table 1. Our analysis of the average fragment numbers and experiments conducted using DGL [8] indicate that the choice between the two sampling strategies has negligible impact on the baseline performance.

For each sampled point $p$ on $P_i$, its fracture label $c_p$ is determined by the distance from $p$ to its nearest neighbor $q$ among points from all other pieces:

$$c_p = \mathbb{1}\left(\min_{q \in O \setminus P_i} \|p - q\|_2 < \eta\right) \quad (15)$$

where $\eta$ is set to 0.02 for all the objects. The ground-truth matching point $\hat{q}$ of a fracture point $\hat{p} \in \hat{P}_i$ is set to

$$\hat{q} = \operatorname*{argmin}_{\hat{q}' \in \hat{O} \setminus \hat{P}_i} \|\hat{p} - \hat{q}'\|_2. \quad (16)$$

where $\hat{O}, \hat{P}_i$ denotes the set of fracture points in $O$ and $P_i$. During the training process, $c_p$ was applied to segment the fracture points and during testing the predicted $\tilde{c}_p$ was used instead. An example is shown in Fig. 7.

Figure 7: An example of surface segmentation over the point cloud of a bottle broken into 7 major pieces. Fracture points are marked with green in the predicted result of our method (left) and red in the ground-truth (right).

After the ground-truth labeling and matching were computed based on the pieces at their original pose, all pieces were recentered to the origin and a random rotation was applied to each piece.

To ensure a fair comparison with baseline methods [26, 10, 27, 8], we adopted the same implementation as of the benchmark code of [4], which sampled the same number of point each piece. For PREDATOR, we applied the same sample routine as the baseline methods, and we paired the adjacent pieces for training and testing.

## C.2 Evaluation Metrics

The mean absolute error MAE(R) and square-rooted mean squared error RMSE(R) of rotation were computed as

$$\text{MAE(R)} = \frac{1}{3}||\tilde{R} - R^{gt}||_1,$$

$$\text{RMSE(R)} = \frac{1}{\sqrt{3}}||\tilde{R} - R^{gt}||_2, \quad (17)$$

Table 4: Comparison of training and influence time with Tesla V100 GPUs. For Jigsaw, the time used in the forward is 1.30s/batch and the rest of the time is used on Hungarian and global alignment.

|  | Jigsaw | Predator[16] | DGL [33] | LSTM [33] | Global [33] |
|---|---|---|---|---|---|
| Training Time / GPUs | 120H / 4 | 96H / 4 | 11H / 1 | 16H /1 | 21H / 1 |
| Influence Speed (s/batch) / batch size | 7.67(1.30) / 8 | 1.28 / 28 | 2.46 / 32 | 1.63 / 32 | 1.65 / 32 |

Table 5: Detailed quantitative results of Jigsaw on Breaking Bad dataset (mean and STD by 3 runs).

| Method | RMSE (R) ↓ degree | MAE (R) ↓ degree | RMSE(T)↓ $\times 10^{-2}$ | MAE (T) ↓ $\times 10^{-2}$ | PA ↑ % |
|---|---|---|---|---|---|
| Results on the `everyday` object subset. | | | | | |
| Jigsaw | $42.3 \pm 0.03$ | $36.3 \pm 0.06$ | $10.7 \pm 0.02$ | $8.7 \pm 0.02$ | $57.3 \pm 0.012$ |
| Results on the `artifact` object subset. | | | | | |
| Jigsaw | $52.4 \pm 0.09$ | $45.4 \pm 0.14$ | $22.2 \pm 0.05$ | $19.3 \pm 0.04$ | $45.6 \pm 0.015$ |

for each piece, where $\tilde{R}$ and $R^{gt}$ were predicted rotation and ground-truth rotation represented in Euler angles. The error over each object was computed as the mean error of all the pieces and the total error was the mean error of all the object. For translation, MAE(T) and RMSE(T) were computed in the same way

$$\text{MAE(T)} = \frac{1}{3}||\tilde{t} - t^{gt}||_1,$$
$$\text{RMSE(T)} = \frac{1}{\sqrt{3}}||\tilde{t} - t^{gt}||_2,$$

$$(18)$$

where $\tilde{t}$ and $t^{gt}$ were the predicted translation and ground-truth translation.

# D  Implementation Details

## D.1  Additional Configuration of parameters

For PREDATOR [16], we carefully follow the parameters presented in its open sourced code, and the threshold for overlap recall to add saliency loss is set to be 0.3. For Jigsaw, we start training with only the segmentation loss. We add matching loss after first 10 epochs, and rigidity loss after 200 epochs.

## D.2  Running time

Our framework is implemented in Pytorch. All methods are trained over Tesla V100-SXM2-32GB GPUs and distributed data parallel strategy is applied for multi-GPU training. The training and influence time for Jigsaw and baseline methods are shown in Table 4.

# E  Additional Results

## E.1  Detailed Quantitative Results

Fig. 8 gives a detailed distribution of each error metric with respect to the number of fractured pieces. Our approach exhibits a clear advantage in each of the metrics evaluated, and this advantage becomes more pronounced as the number of pieces decreases.

In addition, we present the mean and standard deviation (STD) of three runs of our method in Table 5. These values demonstrate the stability of our method across different random seeds and variations in the sampled point cloud. The consistent performance and low standard deviation affirm the robustness of our proposed approach.

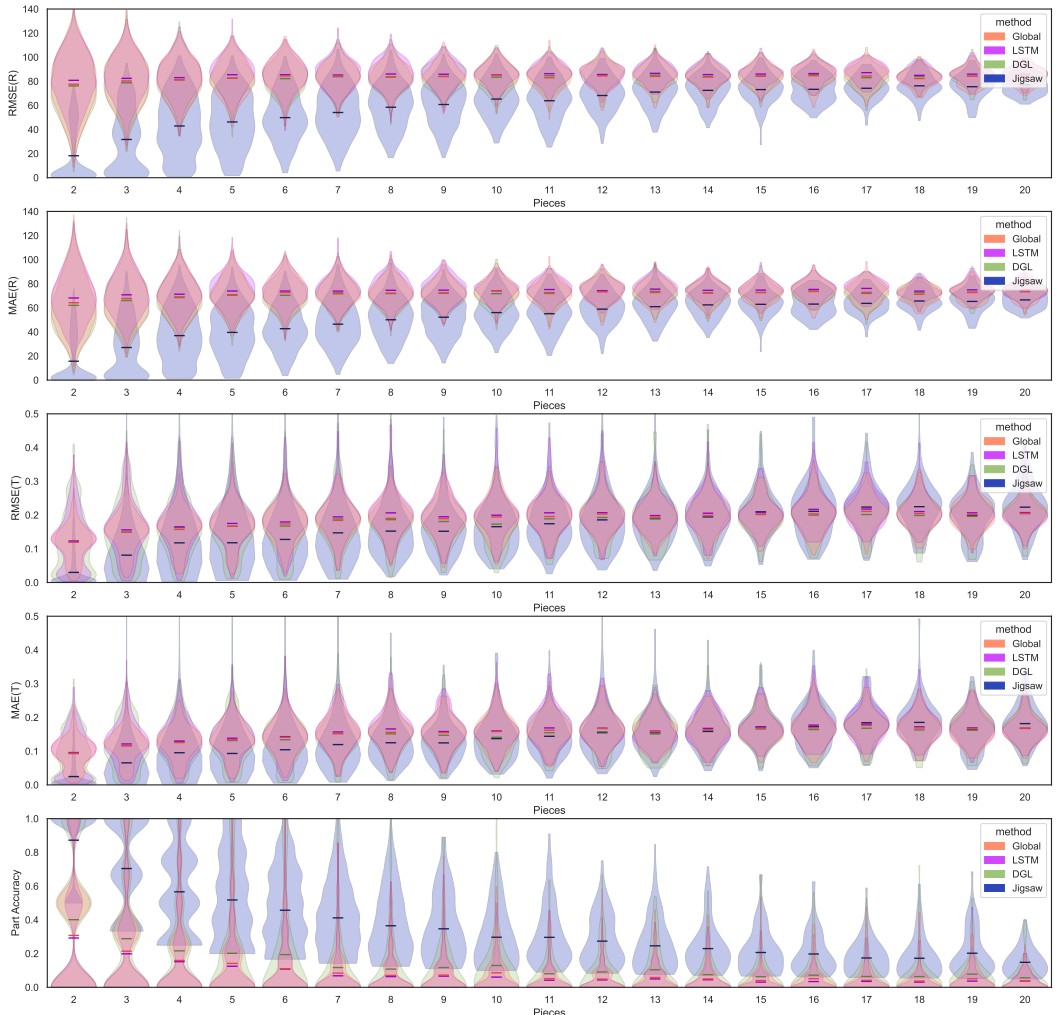

Figure 8: Detailed analysis of the quantitative results obtained by each method on the `everyday` object subset. We provide results for each metric, categorized by the number of pieces involved. The shadowed area in the figures represents the distribution of each metric across the corresponding number of pieces. The horizontal line indicates the mean value of the metric for that specific number of pieces. Different hues represent different methods.

## E.2 More Qualitative Results

We collect additional visualizations of the assembled objects in Fig. 9. Even with a large number of pieces to assemble, our method remains robust in restoring major pieces to their original pose.

## E.3 Additional Registration Baseline

We have received advice to consider GeoTransformer [30] as a state-of-the-art baseline for low overlap registration. However, training GeoTransformer proves to be exceedingly slow (about 15 days over 4 V100 GPUs), primarily due to the significant scale of Breaking-Bad in comparison to the 3D registration datasets on which it was originally tested. On pairwise transformations GeoTransformer achieves MAE(R)=72.4(degree), RMSE(R)=84.8(degree), RMSE(T)=14.3($\times 10^{-2}$), MAE(T)=11.6($\times 10^{-2}$), PA=3.1%. These findings align with the claims we've put forth in our paper, and we believe that presenting the full results of PREDATOR adequately reflects how registration baselines perform on the assembly task.

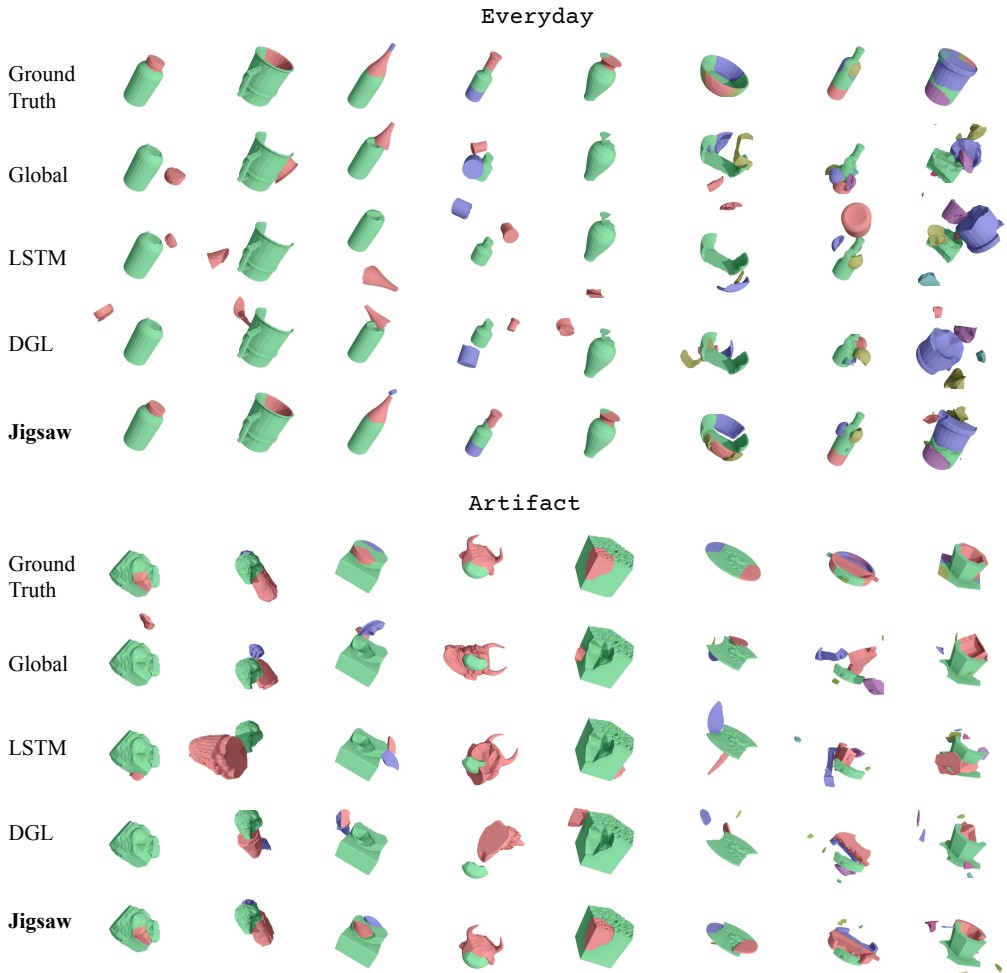

Figure 9: More visualization of assembly results.

