# OpenReview forum: "Jigsaw: Learning to Assemble Multiple Fractured Objects"
_NeurIPS.cc/2023/Conference — NeurIPS 2023 poster_

### Official Review · Reviewer_nZZN · 2023-06-26

**Soundness:** 3 good
**Presentation:** 4 excellent
**Contribution:** 3 good
**Rating:** 7
**Confidence:** 5

**Summary:**

The authors address the problem of reassembling fractured 3D objects from its pieces, where each piece is represented as a point cloud. The main challenge lies in the fact that one has to simultaneously solve a segmentation problem (which part of an object is fracture surface and which part is the original surface) and a multi-matching problem. The authors propose a learning-based approach that nicely take into account these different aspects. Their approach is supervised, which sounds reasonable given that it is relatively straightforward to simulate object fractures from complete objects.

**Strengths:**

+ The authors address a difficult, relevant and relatively novel problem and set the new state of the art
+ The proposed solution is a neat combination of learning and traditional optimisation techniques
+ All pieces are matched simultaneously which really is key to address this difficult problem
+ The idea of using a primal-dual feature descriptor pair to account for the complementary geometric properties of features is novel and useful

**Weaknesses:**

- Although the presented results look appealing, they only show a small subset of all results. Given the relatively large angular errors (up to 52.4 degrees), it is unclear whether results are cherry-picked or representative. Given the overall improvement over previous methods I do not see this as critical for acceptance, yet, it is important to understand this.
- There a few things that leave some room for clarification (see below).

**Questions:**

1) The intro also discusses furniture assembly - however, this is quite a different problem since furniture does not have fractures. I suggest to smoothen this.
2) In related work it is mentioned that most graph matching methods focus on pairwise matchings. There are some recent multi-matching methods which should be discussed (see e.g. Universe Points Representation Learning for Partial Multi-Graph Matching, AAAI'23 and references therein).
3) Please make the font and visuals in all figures larger (ideally the fonts match the font size of the caption).
4) How many V100 GPUs are in one machine? Please comment on the training and inference time.
5) I think there is some ambiguity in the notation around lines 161-165: $N$ is already introduced before as the number of object pieces; also, the index $i$ is used both for the object piece index, and as point index.
6) How do you ensure that the primal-dual feature pair is really complementary/consistent? Is this enforced by construction or does the learning take care of that? Also, the formulation "the surface looks different from a different viewpoint" is not ideal, since this suggests there is some camera (with projection) involved (at least that was my first thought). I suggest to rephrase this.
7) Is matrix $A$ in eq. (6) positive semidefinite? It looks a bit like the Mahalanobis distance in a Gaussian kernel, but then we would also need a negative sign. Please comment on this.
8) Eqn. (9) misses that (R,t) are elements of SE(3). Similarly, the solution below finds (R,t) in E(3) - one needs to ensure the determinant of the rotation is +1.
9) How do you ensure that the blocks of the matrix X in eqn. (13) are consistent? In 3.4 it is explained that pairwise transformations are computed first (representing blocks of X), and then RANSAC is applied. Does this guarantee global consistency? Do you apply some variant of permutation synchronisation, or is consistency only guaranteed for the rigid-body transformations via the Shonan-averaging?
10) What is the scale/unit of the point clouds? Absolute values of transformation error are not very meaningful.

**Limitations:**

Limitations are adequately addressed.

---

> ### Author Rebuttal · Authors · 2023-08-09
>
> Thanks for your time and valuable comments. Thank you for your suggestions regarding the introduction, multi-graph matching references, and figure enhancements. We will make adjustments in the final version.
>
> Below we respond to your questions and hope we can clarify the potential misunderstandings.
>
> > Although the presented results look appealing, they only show a small subset of all results. Given the relatively large angular errors (up to 52.4 degrees), it is unclear whether results are cherry-picked or representative. Given the overall improvement over previous methods I do not see this as critical for acceptance, yet, it is important to understand this.
>
> **A:** In Figure 8 we showed the error distribution with respect to the number of pieces. While we achieved general superiority, there remains space for improvement especially for over 5 pieces. We issued this limitation in Section 5 and left it as our future work.
>
> > Q4. How many V100 GPUs are in one machine? Please comment on the training and inference time.
>
> **A:** 4 V100 GPUs are used in one machine. Please refer to the general response for detailed information on time and resource allocation.
>
> > Q5. I think there is some ambiguity in the notation around lines 161-165:
>  is already introduced before as the number of object pieces; also, the index
>  is used both for the object piece index, and as point index.
>
> **A:** $n$ should be used to represent the number of object pieces and $N$ represents the number of points. We will adjust them accordingly in the final version.
>
> > Q6. How do you ensure that the primal-dual feature pair is really complementary/consistent? Is this enforced by construction or does the learning take care of that? Also, the formulation "the surface looks different from a different viewpoint" is not ideal, since this suggests there is some camera (with projection) involved (at least that was my first thought). I suggest to rephrase this.
>
> **A:** The affinity metric's design and the learning process, guided by the matching loss, jointly ensure that the primal-dual feature pair exhibits a preference for matching points with complementary geometry rather than similar geometry. A more detailed explanation is presented in the general response and we will further clarify this aspect in the final version.
>
>
> > Q7. Is matrix  in eq. (6) positive semidefinite? It looks a bit like the Mahalanobis distance in a Gaussian kernel, but then we would also need a negative sign. Please comment on this.
>
> **A:** In practice, Matrix A contains learnable parameters to weight point features and no PSD constraints are explicitly applied. This is a common practice in matching tasks since the paper "Deep Learning of Graph Matching", A Zanfir et. al, CVPR 2018.
>
> > Q8. Eqn. (9) misses that (R,t) are elements of SE(3). Similarly, the solution below finds (R,t) in E(3) - one needs to ensure the determinant of the rotation is +1.
>
> **A:** If the determinant of R is -1, we flip the singular vector with the smallest singular value in V.
>
> > Q9.1. How do you ensure that the blocks of the matrix X in eqn. (13) are consistent?
>
> **A:** All surface points are matched collectively and then organized to create pairwise matching. Thus, there's no consistency concern regarding the matrix X computed from Eq (13).
>
> > Q9.2. In 3.4 it is explained that pairwise transformations are computed first (representing blocks of X), and then RANSAC is applied. Does this guarantee global consistency?
>
> **A:** RANSAC was applied for computing pairwise transformation from point correspondences. For consistent global pose estimation we applied Shonan-averaging, which was a state-of-the-art robust alignment method.
>
> > Q9.3 Do you apply some variant of permutation synchronisation, or is consistency only guaranteed for the rigid-body transformations via the Shonan-averaging?
>
> **A:** In the fracture assembly task each fracture point is supposed to have one unique corresponding point among all other pieces. Therefore the permutation synchronization issue is not involved. Moreover, this global one-to-one matching was a critical insight for our multi-part matching design.
>
> > Q10. What is the scale/unit of the point clouds? Absolute values of transformation error are not very meaningful.
>
> **A:** Each object has a scale of ~0.8 on average in each dimension, while each piece is approximately ~0.3 in each dimension. It's important to note that translation errors can accumulate for matching-based methods if a piece is incorrectly assembled to a wrong position.

---

> > ### Comment · Reviewer_nZZN · 2023-08-11
> >
> > Thank you for the detailed comments. I am assuming that, although not explicitly mentioned, you will clarify all mentioned aspects in the final version. With that I will keep my initial rating.

---

### Official Review · Reviewer_jmgU · 2023-07-05

**Soundness:** 2 fair
**Presentation:** 3 good
**Contribution:** 2 fair
**Rating:** 4
**Confidence:** 3

**Summary:**

This paper proposes a novel framework for the task of 3D fracture assembly. The proposed approach leverages hierarchical features of global and local geometry to match and align the fracture surfaces. Experiments are conducted on the Breaking Bad dataset to verify the effectiveness of the proposed method.

**Strengths:**

1. This paper is well-written and easy to follow;

**Weaknesses:**

1. I am not the guy from the same area and therefore I cannot judge the novelty of the proposed mehtod well. However, from my experiences in my research field, this task is quite close to pairwise and multi-frame point cloud registration, and the major difference is the utilization of surface segmentation and the significantly lower overlap ratio between parts. Based on that, some techniques, especially the geometric learning and the matching, are well-explored and widely-adopted in the aforementioned registration tasks, and therefore I consider the proposed solution trivial.

2. For the experiments, I think there are some problems:

   2.1 The experiments are conducted on a single dataset, and I am not sure whether it is a synthetic or real one. To this end, I think it is necessary to also evaluate the proposed method on a second dataset. If the leveraged dataset is synthetic, it would be better to include experiments on real data;

    2.2 In Tab. 1, the latest baselines are from 2020 which is 3 years ago. I have no idea why some latest ones are not included. Does it mean that there have been no work in this topic for 3 years? I would like to see the comparisons to some more recent works during rebuttal;

    2.3 In Tab.1, the only point cloud registration baseline is Predator which is a work in 2021. Some more recent works, for example, CoFiNet (NeurIPS 2021), GeoTransformer (CVPR 2022), Lepard (CVPR 2022), RegTr (CVPR 2022) should be considered as the baselines from the field of point cloud registration (GeoTransformer is the state-of-the-art).

    2.4 As the task is similar to multi-frame point cloud registration, I also think it reasonable and necessary to include baselines from that area, like (1) and (2).

3. Fig. 1 carries overly much information and is hard to focus on each specific part. I would suggest the authors modifying it to make it clear. Also, in Related Work (line 90), [30] is not based on the overlap prediction.


---------------------------------------------------------------------------------------------------------

(1) Gojcic et al. Learning Multiview 3D Point Cloud Registration, CVPR 2020

(2) Yew et al. Learning Iterative Robust Transformation Synchronization, 3DV 2021

**Questions:**

1. As in the cross-attention, all the points from all the parts are leveraged for the attention computation. Therefore, the efficiency heavily relies on the size of point clouds. So how many points are included in each part and the whole shape? Is the proposed method still applicable if the number of points raised to >10k?

2. What is the main difference between the assembly task and the task of pairwise/multi-frame point cloud registration?

3. What is the overlap ratio (probably in average) between different parts?

**Limitations:**

Limitations have been discussed in the main paper.

---

> ### Author Rebuttal · Authors · 2023-08-09
>
> Thanks for your time and valuable comments. Below we respond to your questions/concerns point-by-point and hope we can resolve some confusions.
>
> > 2.1 The experiments are conducted on a single dataset, and I am not sure whether it is a synthetic or real one. To this end, I think it is necessary to also evaluate the proposed method on a second dataset. If the leveraged dataset is synthetic, it would be better to include experiments on real data;
>
> **A:** Breaking Bad is a synthetic dataset introduced in NeurIPS 2022, and is the only available option for multi-part fracture assembly assessment. It already exhibits considerable size and diversity, particularly regarding its fracture modes and object categories. As fracture assembly remains a relatively new field, we believe efforts towards introducing a distinct dataset for evaluation are underway.
>
> > 2.2 In Tab. 1, the latest baselines are from 2020 which is 3 years ago. I have no idea why some latest ones are not included. Does it mean that there have been no work in this topic for 3 years? I would like to see the comparisons to some more recent works during rebuttal;
>
> **A:** Fracture assembly remains a relatively new field, progressing at a gradual pace. Based on the Breaking Bad Dataset publication in NeurIPS 2022, the latest relevant baseline listed is DGL. We have not encountered any new publications pertaining to this topic in the recent half-year period.
>
> > 2.3 In Tab.1, the only point cloud registration baseline is Predator which is a work in 2021. Some more recent works, for example, CoFiNet (NeurIPS 2021), GeoTransformer (CVPR 2022), Lepard (CVPR 2022), RegTr (CVPR 2022) should be considered as the baselines from the field of point cloud registration (GeoTransformer is the state-of-the-art).
>
> **A:** GeoTransformer was indeed considered as a registration baseline. However, after preliminary experiments, its performance aligned with that of Predator, rendering it unsuitable for providing meaningful assembly results. Given the unaffordable time investment required for GeoTransformer's training (estimated 15+ days), it was not included in our results.
>
> > 2.4 As the task is similar to multi-frame point cloud registration, I also think it reasonable and necessary to include baselines from that area, like (1) and (2).
>
> **A**: Our task is different in the sense that besides we have to segment the fractured surfaces out, each point of the fractured surface has at most one potential match with fractured regions of other fragments. Our approach explicitly leverages this property. This is very different from multi-frame point cloud registration, in which one point may be matched with multiple points (each point from each overlapping scan). Therefore, we do not think it is necessary to compare against multi-frame point cloud registration baselines.
>
> > Q1. As in the cross-attention, all the points from all the parts are leveraged for the attention computation. Therefore, the efficiency heavily relies on the size of point clouds. So how many points are included in each part and the whole shape? Is the proposed method still applicable if the number of points raised to >10k?
>
> **A:** The overall shape contains 5K points, while individual parts’ point numbers are sampled based on the part surface area. We tested that our method is runnable with 10K points sampled in a single object on V100. We acknowledge the possibility of designing downsampling schemes for cross-attention, while this lies beyond the scope for this paper and merits exploration in future endeavors.
>
> > Q2. What is the main difference between the assembly task and the task of pairwise/multi-frame point cloud registration?
>
> **A:** The essential difference is that the assembly task focuses on the thin surfaces and edges of the object and relies on the complementary geometry between pieces. Though it may be perceived as an extreme case of pairwise/multi-frame point cloud registration, key differences include: 1) The overlapping area appears at most in 2 parts (leading to our multi-part matching strategy); 2) The overlap lacks “volume” and should based on complementary geometry instead of finding identical geometry, which requires distinct representation from objects/scenes registration (as demonstrated in our primal-dual descriptor design) 3) Constraints exist for maintaining smoothness in the assembled object's original surface, a matter addressed within our limitations and reserved for future exploration.
>
> > Q3. What is the overlap ratio (probably in average) between different parts?
>
> **A:** On average, the fracture points account for 20.0% of all points. For extreme cases (e.g., a fractured plate) the ratio can fall to 4%. Since each fracture point belongs to a unique match, this ratio indicates the average overlap between pieces.

---

> > ### Comment · Reviewer_jmgU · 2023-08-17
> > **Reviewer Feedback**
> >
> > Dear authors,
> >
> > Thanks for your feedback and I really appreciate your effort. However, I still have questions on some specific aspects:
> >
> > 1. Regarding Q 2.3, GeoTransformer is a strong basline that has much better performance compared with Predator for the point cloud registration task, while the authors claimed that  "its performance aligned with that of Predator". Moreover, the author also said the training of GeoTransformer on their target dataset is unaffordable. Then, how was the conclusion "its performance aligned with that of Predator" drawn?
> >
> > 2. Why is the training of GeoTransformer on the target dataset way slower than that on 3DMatch/3DLoMatch (GeoTransformer's target benchmarks)? From my understanding, your data is synthetic and with fewer points. Does it have more training cases? If yes, then comparing the proposed method with GeoTransformer on 3DMatch/3DLoMatch should be appliable. Considering the similarity of the two tasks, I think this experiment should be included.
> >
> > 3. Regarding Q 2.4, I think from your explanation, the assembly task is an easier version of the multi-frame registration. Therefore it would make sense to apply some of the state-of-the-art multi-frame registration methods to your task as additional baselines.
> >
> > 4. Considering that only a synthetic dataset is used in this paper and the also the similarity between the assembly task and point cloud registration, I would like to see comparing the proposed methods with the registration baselines, e.g., GeoTransformer, on 3DMatch/3DLoMatch to further validate it. If GeoTransformer requires 15days to be trained on your data, I think this experiment is suitable to conduct and affordable to your GPU.
> >
> > I have carefully read other reviewers' comments and found I am the only guy that leans towards the negative side. However, after reading authors' feedback, let me be honest, I have a feeling like the authors tried to find excuses to avoid the comparison with the registration baselines. Also, there are some statements from the authors that I still disagree. According to the previous comments, I will insist my evaluation on this paper. If I was wrong, please correct me. I will reconsider my score afterwards.
> >
> > Best,
> >
> > Reviewer jmgU

---

> > > ### Author Response · Authors · 2023-08-19
> > >
> > > Thanks for your comments. We think these discussions will help to make our paper better.  Let us try to answer your questions from another perspective which we think may align with your perception of all relevant problems.
> > >
> > > First of all, regarding rigid matching. For pairwise matching, all existing techniques are based on learning feature matching and integrating the rigidity constraints. Multiple-scan alignment further enforces the cycle-consistency constraint among pairwise matches. As you have questioned, we do not claim we make contributions on these aspects.
> > >
> > > The key contribution of our Jigsaw paper, as mentioned by other reviewers, is introducing a new formulation for multi-scan alignment. We introduce the problem of joint shape matching and shape segmentation, integrating the uniqueness constraint of fractured surface matching, and develop a novel primal-dual descriptor for fractured-surface matching.  We believe those contributions are sufficient from the technical aspect. They also stimulate future work that develop better machine learning models to integrate these constraints to further improve the performance.
> > >
> > > Now going back your question is applying existing 3D rigid matching approaches on the fractured surface assembly problem. When working on this problem, we started with experimenting with some off-the-shelf rigid matching approaches and did not get any reasonable results. During the rebuttal period we tried GeoTransformer. The training is extremely slow since Breaking-Bad is much larger than both of the proposed 3D registration datasets. The training loss decreases only a bit. We are waiting to see what will happen in another 10 days, but the trend from what people typically do in deep learning is to believe it does not work. As for a preliminary result, on pairwise transformations GeoTransformer achieves MAE(R)=72.4(Deg), RMSE(R)=84.8(DEG), RMSE(T)=14.3(x1e-2), MAE(T)=11.6(x1e-2), PA=3.1%. The average rotation error is just slightly below random guessing. We also tried “Learning Multiview 3D Point Cloud Registration” from CVPR 2020, which is a state-of-the-art multi-view registration approach, on our primal-dual descriptor. The mean rotation error is also around 81.2(Deg). What we have observed from these experiments is that the fractured surface regions are relatively small, making it very difficult for existing approaches to learn any meaningful features. Because of that, they do not work.
> > >
> > > We do not agree that fractured object assembly is easier than scan registration. It is much harder without detecting fractured surfaces. It is also a very different problem because features learned on fractured surfaces also carry neighboring information from original surface regions. This is achieved by learning feature matching and segmentation together. Without performing segmentation and matching together, off-the-shelf registration does not work well. Similarly, we can also argue in scan registration one can utilize priors about the underlying complete object/scene. Similar prior approaches, e.g., “Extreme relative pose estimation for rgb-d scans via scene completion (CVPR 2019)”, have leveraged this insight to perform geometry completion and matched completed scans. This approach has improved the accuracy of relative pose estimation substantially. This is harder to do in the fractured object assembly setting because we have to identify original surfaces and fracture surfaces. In our Jigsaw approach, this is partially addressed in the primal-dual descriptor which captures information from original surfaces. According to your rationale, we can also say scan registration is easier than fractured object reassembly.
> > >
> > > We hope these comments help you understand our points better.

---

> > > > ### Comment · Reviewer_jmgU · 2023-08-19
> > > >
> > > > Thanks for the detailed explanation. Now I understand the task much better and acknowledg the value of solving the assembly task through the proposed method.
> > > >
> > > > However, regarding my previous Q4 and also the review from Reviewer Pfa7, I still consider it unavoidable to compare with registration methods on their real data benchmarks (because of the relavence of the two tasks and also since your paper lacks real data evaluation), for better demonstrating the value of your method.
> > > >
> > > > Now I am raising my score a little bit and looking forward to your feedback.
> > > >
> > > > Best,
> > > >
> > > > Reviewer jmgU

---

> > > > > ### Author Response · Authors · 2023-08-21
> > > > >
> > > > > We appreciate your additional comments. Our approach builds upon two principles that differentiate the assembly and registration problem:
> > > > >
> > > > > 1. Complementary fracture surfaces: assembly focuses on pairing complementary fracture surfaces.
> > > > > 2. One unique matching for each point: assembly requires that each fracture point is matched uniquely to one counterpart.
> > > > >
> > > > > In contrast, registration involves identifying overlapping regions that share structural identity, not complementarity. This contrasts with our first principle. Moreover, the one unique matching rule is not applicable in the multi-frame registration scenarios, where a point can appear in more than three frames.
> > > > >
> > > > > These tailored principles make our current pipeline incompatible with registration problems. Applying our method to the registration problem is way beyond the scope of this paper, and we therefore leave this exploration as a possible future work.
> > > > >
> > > > > We acknowledge the absence of a comprehensive real-world dataset for multiple fracture assembly and we sincerely anticipate the opportunity to validate our method if such data emerges. As a point of clarification, “Neural Shape Mating” mentioned by Reviewer Pfa7 is a synthetic dataset for two pieces’ assembly.

---

> > > > > > ### Author Response · Authors · 2023-08-21
> > > > > >
> > > > > > We hope to further explain our perspective on using the synthetic dataset.
> > > > > >
> > > > > > As we have stated, we do agree that eventually, one has to develop and evaluate an algorithm on real datasets. However, for the fractured assembly problem, we think this is not necessary at this moment from the perspective of advancing the field. The synthetic dataset is already very challenging. Even with our approach, the average mean rotation error is around 40 degrees. We believe that this synthetic data, which was introduced last year at NeurIPS will continue to serve as the benchmark for developing better fractured object reassembly algorithms. When an algorithm eventually solves the problem on this dataset (e.g., less than 10 degrees), we are ready to move into the real data regime. However, along the way we believe as a community we will develop much better algorithms and understand the problem better. Also, obtaining a real fractured object dataset requires a lot of work.
> > > > > >
> > > > > > As an example from a similar field, the ShapeNet dataset consists of just synthetic 3D objects. Many algorithms were developed and are still being developed and evaluated on this benchmark dataset. Later people introduced real datasets, e.g., ScanObjectNN, for experimental evaluation. Many algorithms focus on leveraging pre-trained models on ShapeNet and fine-tuning them on ScanObjectNN. We think eventually the community will do something similar, but we are just not at that stage yet.
> > > > > >
> > > > > > We hope this comment may clarify our points.

---

### Official Review · Reviewer_qeDQ · 2023-07-06

**Soundness:** 3 good
**Presentation:** 3 good
**Contribution:** 3 good
**Rating:** 6
**Confidence:** 3

**Summary:**

The authors present a learning framework for assembling physically broken 3D objects with point cloud observations. There are three components in the framework, surface segmentation to identify the fracture points, multi-parts matching to find correspondences among the fracture points, and the global alignment to recover the global poses of the pieces. The main contributions are the joint learning of segmentation and multi-parts matching and the design of the multi-parts matching including the primal-dual descriptor and the losses. The contributions are validated by superior performance on the Breaking Bad dataset compared with other methods.

**Strengths:**

The task is new with just few methods already developed. This work may serve as a baseline for future work in this line.

The overall framework is clearly clarified and organised. The adding of self-attention and cross-attention is well justified and illustrated for the front-end feature extractor. The primal-dual descriptor is introduced in the multi-part matching part, which is new for this task to my knowledge. A further analysis of the primal-dual descriptor is also provided in the appendix to provide better understanding of the effect.

The experiments and results are also solid.

**Weaknesses:**

The absence of the statistics of the number of pieces in the dataset may lead to unfair comparison against the baseline methods as points are sampled at different level as shown in Table 3 in the appendix.

A further analysis of the multi-parts matching w.r.t. the number of pieces could better show the effect of multi-part matching against pairwise matching.

The notations are sometimes a bit confused, especially for the tilde and hat. For example, the hat_P_i at line 258 seems to be not defined.

The experimental settings are not clear for all the involved methods. A table to list and compare the conditions and data/label requirement for each method would be helpful.

The introduction of the rigidity loss is not well justified and analyzed, especially it is only applied to the model training after 200 epochs.

Texts in the figures are too small to recognize.

**Questions:**

Does the generation of the segmentation labels require additional knowledge about the data compared to other methods? Although I don't think so, a clarification from the authors would be helpful.

Is only one precise match at line 193 an assumption? The assumption of only one precise match may not be satisfied as the points sampled on neighboring pieces may be quite far and no matching can be done simply based on point clouds.

Addressing the weaknesses listed above could change my opinion.

**Limitations:**

The authors deals with the accuracy drop of pose estimation when the number of fractured pieces increases.

---

> ### Author Rebuttal · Authors · 2023-08-09
>
> Thanks for your time and valuable comments. Below we respond to your concerns and hope we can clarify some questions.
>
> > The absence of the statistics of the number of pieces in the dataset may lead to unfair comparison against the baseline methods as points are sampled at different levels as shown in Table 3 in the appendix.
>
> **A:** We include the distribution of the number of pieces in Figure 2 of the authors' response document. The median number of fractured pieces is $4$ and the mean is $6.1$. Given the different backbone structure of different methods, it is infeasible to use the same sampling strategy among all the methods. To ensure equitable assessment, we therefore uniformly sampled 5000 points from each object in our method, akin to a 5-piece scenario for baseline methods. We additionally provide a detailed error distribution of our method across all potential piece numbers in Fig. 8 of appendix. This visual clearly represents our method's superiority across all subset variations.
>
> > A further analysis of the multi-parts matching w.r.t. the number of pieces could better show the effect of multi-part matching against pairwise matching.
>
> **A:**
> Figure 8 in the appendix provides an in-depth comparison of performance concerning the number of pieces in objects.
>
> > The notations are sometimes a bit confused, especially for the tilde and hat. For example, the hat_P_i at line 258 seems to be not defined.
>
> **A:**
> Our notation convention assigns a hat to variables associated with fractures and a tilde to variables denoting predictions or approximations. Specifically, $\hat{P_i}$ represents the fracture points on piece $i$, and we will introduce a clarification for this notation.
>
> > The experimental settings are not clear for all the involved methods. A table to list and compare the conditions and data/label requirement for each method would be helpful.
>
> **A:** We include the experimental settings in Table 3 in the appendix. For all baseline methods and our approach, only the ground-truth poses are required for supervision.
>
> > The introduction of the rigidity loss is not well justified and analyzed, especially it is only applied to the model training after 200 epochs.
>
> **A:**
> The rigidity loss increased the performance by $0.2 (\times 10^{-2})$ in translation. It serves as a regularization to encourage rigid dense point matching, and its application requires a well-informed initialization. To meet this requirement, we train 200 epochs before introducing the rigidity loss.
>
> > Does the generation of the segmentation labels require additional knowledge about the data compared to other methods?
>
> **A:**
> No additional knowledge about the data is required. Segmentation labels are computed based on the original Breaking Bad dataset's ground truth position of each piece (lines 171-173). This process doesn't involve any additional manual labor or introduce new information into the dataset.
>
> > Is only one precise match at line 193 an assumption? The assumption of only one precise match may not be satisfied as the points sampled on neighboring pieces may be quite far and no matching can be done simply based on point clouds.
>
> **A:** This is an observation that aligns with the real-world perspective when we treat surfaces as continuous. While sampled point clouds might not perfectly adhere to this rule due to the approximations introduced, the underlying observation guides our algorithm design of multi-part matching.

---

> > ### Comment · Reviewer_qeDQ · 2023-08-19
> >
> > I thank the author for the detailed response. The response addresses part of my concerns. But the following questions are still not very clear to me.
> >
> > **Sampling.** I am still not convinced that the setting of the sampling strategies are fair comparison.
> >
> > **Experimental settings.** While I am aware the presence of Table 3, I actually mean the dataset preprocessing for each method.
> >
> > **One precise match assumption.** I understand this assumption could guide the design of the algorithm. However, since this assumption is not always satisfied, it is worth to discuss or show the consequence when it is not satisfied.

---

> > > ### Author Response · Authors · 2023-08-20
> > >
> > > Thanks for your comment. We hope by providing the following results and statistics, we can answer your questions.
> > >
> > > **Sampling.** During the rebuttal period, we conducted an experiment employing the same sampling strategy as our method on DGL. The result (5000/obj) compared to the reported baseline (1000/p) is as follows:
> > > |        | RMSE(R) | MAE(R) | RMSE(T) | MAE(T) |   PA   |
> > > |:--------|:---------:|:--------:|:---------:|:--------:|:--------:|
> > > |  DGL (1000/p)   |  80.6   |  67.8  |  15.8   |  12.5  |  23.9  |
> > > | DGL (5000/obj)  |  81.1  | 68.1  |   15.4  |  12.3  |  25.5  |
> > >
> > > It's evident from the results that the change in sampling strategy has had no impact on the outcome. Other assembly baselines should have similar results as they share similar building blocks as DGL. We hope this addresses any concerns regarding the sampling strategies.
> > >
> > > **Dataset setting.** The only difference in the dataset processing lies in the sampling strategy. We will expand Table 3 regarding these settings in the final version.
> > >
> > > **One precise match assumption.** Regarding the assumption of one precise matching, our analysis indicates that, in the ground truth, approximately 80% of fracture points can be satisfied with the one precise match assumption. We believe this proportion is sufficient for computing the final pose. Furthermore, our global alignment technique utilizing RANSAC is robust enough to accommodate these minor mismatches. Therefore, the consequences of this assumption not being strictly met are minor and manageable.

---

> > > > ### Comment · Reviewer_qeDQ · 2023-08-21
> > > >
> > > > Thanks for the comment.
> > > >
> > > > The sampling and dataset setting are clear to me now.
> > > >
> > > > **One precise match assumption.** I am not trying to criticize that the one precise match assumption is not always satisfied. I point out this to see what is the impact when the assumption is violated and how it can be managed.

---

### Official Review · Reviewer_Pfa7 · 2023-07-07

**Soundness:** 4 excellent
**Presentation:** 3 good
**Contribution:** 4 excellent
**Rating:** 5
**Confidence:** 5

**Summary:**

This paper proposes Jigsaw, a novel framework to assemble 3D fragments of a shape to complete the whole 3D shape.
Jigsaw first segments the fracture surface, since only the points on the fracture surface are involved in the physical assemble between fragments. Jigsaw then performs multi-part matching to find correspondences among the fracture surface points, which is finally used to recover the global pose of each fragment via global alignment.
Jigsaw exhibits state-of-the-art performance on the Breaking Bad dataset.
The authors propose Jigsaw to be the first learning-based method to assemble 3D fractures over **multiple** pieces.

**Strengths:**

* First learning based method to tackle the task of multi-part assembly of 3D fragments.

* Well-motivated, novel learning pipeline to facilitate multi-part assembly of 3D fragments.

* Strong performances on the Breaking Bad dataset. Considering that a two-fragment assembly problem can also be seen as a zero-overlap (or surface-overlap) point cloud registration, the baseline methods in the table are also appropriate.



**Weaknesses:**

* Lacks self-containedness. While it is understandable that Shonan-averaging was used as the off-the-shelf global alignment method, but the paper does not go on to explain how this global alignment functions at the minimum, nor how the global pose values from Shonan-averaging is used for evaluation against the given ground-truth global pose values.

* Jigsaw samples a fixed number of points across all the fragments of a single shape. In that case, Jigsaw is expected to fail given a particularly small fragment, from which a very small number of fragments would be sampled. This may result in not a single point on the fracture surface being sampled, and this lacks emphasis in the manuscript.

* The manuscript lacks explanation on how the primal-dual descriptor works. The authors state that their 'system learns surface features that capture the characteristics of a local surface from both directions' - which lacks the evidence the substantiate this claim. Is it the **aim** of primal-dual descriptors to function as mentioned, or is there a theoretical evidence to prove that the primal-dual descriptors actually captures such characteristics?

* While the idea of surface segmentation to narrow down the points to consider is viable, it is expected that the fracture surface is rather easy to segment due to its irregularity in comparison to other parts of the shape. How would the proposed method perform if the fracture is not irregular, but of regular cuts (e.g., straight lines, smooth curves) such as in Neural Shape Mating? [1]

[1] YC Chen et al., Neural Shape Mating: Self-Supervised Object Assembly with Adversarial Shape Priors, CVPR 2022

**Questions:**

Please refer to the above weaknesses section. It would be appreciated if the authors could include additional details to improve the self-containedness of the paper, and to clear any potential misunderstandings.

**Limitations:**

The limitations of Jigsaw have been discussed by the authors.

---

> ### Author Rebuttal · Authors · 2023-08-09
>
> Thanks for your time and valuable comments. Below we respond to your comments and hope we can clear the potential misunderstandings.
>
>
> > Lacks self-containedness. While it is understandable that Shonan-averaging was used as the off-the-shelf global alignment method, but the paper does not go on to explain how this global alignment functions at the minimum, nor how the global pose values from Shonan-averaging is used for evaluation against the given ground-truth global pose values.
>
> **A:** Shonan-averaging computes a global alignment based on pairwise transformations and outputs a transformation matrix that implies a global pose within a specific global coordinate system. The input organization for Shonan-averaging is detailed in Section 3.4. For assessing against the ground truth, we adopt the canonical coordinate system of the largest piece, termed the anchor piece, as the global frame of reference (i.e., the global pose of the largest piece is consistently set to identity), and then align all other pieces relatively. The error is quantified by contrasting the predicted and ground-truth relative transformations of all other pieces with respect to the anchor. We will incorporate this comparison procedure into section 3.4 for clarification.
>
> >  ... may result in not a single point on the fracture surface being sampled.
>
> **A:** We assure that each fragment is sampled with a minimum of 30 points to ensure compactness. We will add this explanation to our manuscript.
>
> > The manuscript lacks explanation on how the primal-dual descriptor works. The authors state that their 'system learns surface features that capture the characteristics of a local surface from both directions' - which lacks the evidence to substantiate this claim. Is it the aim of primal-dual descriptors to function as mentioned, or is there theoretical evidence to prove that the primal-dual descriptors actually capture such characteristics?
>
> **A:** In Figure 6, we show a comparative analysis between the primal-dual descriptor and the
> conventional single-view descriptor. Experimental results show that the primal-dual descriptor highly favored matches between fracture points with complementary local geometries compared to identical local geometries. Please refer to the general response for a more detailed explanation.
>
>
> > While the idea of surface segmentation to narrow down the points to consider is viable, it is expected that the fracture surface is rather easy to segment due to its irregularity in comparison to other parts of the shape. How would the proposed method perform if the fracture is not irregular, but of regular cuts (e.g., straight lines, smooth curves) such as in Neural Shape Mating? [1]
>
> **A:** Our framework is explicitly designed for fracture assembly scenarios involving objects fractured due to physical forces. The Breaking Bad dataset is generated through physical simulation where the typical fracture pattern is irregular and rugged. In instances where a smoother fracture pattern is introduced, we believe that prior knowledge of the object’s shape is necessary. A hypothetical scenario could involve cutting a pyramid into two tetrahedrons. In the absence of shape priors, it would be reasonable to match the base or sides of the two fragments together.

---

> > ### Comment · Reviewer_Pfa7 · 2023-08-19
> >
> > Thank you for the detailed response to my concerns. Regarding the four weaknesses I have written in my review, I believe that the first three has been adequately answered and that the authors will include these details in the final manuscript (or in the appendix) to clear any potential misunderstandings of the readers.
> >
> > However, regarding the fourth weakness (..Neural Shape Mating? [1]), the authors' response proposes what could be 'reasonable', while my question was 'how would the proposed method perform'. I believe that smooth fracture patterns are also realistic (and that the Breaking bad dataset fails to capture this aspect), and I still think that the proposed surface segmentation scheme is effective largely due to the irregularity of the fractured surface. I would appreciate it if the authors could conduct a simple experiment regarding this aspect, or provide an opinion on 'how the proposed method would perform'.
> >
> > I have read through the other reviews, and agree with the particular concern that the paper currently lacks 'rigorous' comparisons compared to existing 'point cloud registration' pipelines. On the other hand, I believe that the task at hand differs from existing point cloud registration problems, as fractured object assembly can be considered as a 'zero-overlap' point cloud registration whereas point cloud registration assumes that there is at least a slightly overlap between the two point clouds. This is where I believe the primal-dual descriptor plays a key role. To conclude, while I agree that the paper could be improved by including additional comparisons with more SoTA point cloud registration methods, the task at hand (assembling fractured objects) is distinct from the task of point cloud registration, and thus the positioning of the submitted paper is appropriate.
> >
> > I would like to maintain my current rating of borderline accept as of now.

---

> > > ### Author Response · Authors · 2023-08-20
> > >
> > > Thanks for your comments.  Regarding concern of the smooth fracture issue, we believe a more versatile fracture segmentation module fine tuned over smooth patterns will augment the capabilities of our method. Our current observations indicate that even in cases where the fracture surface exhibits relative smoothness, our segmentation module adeptly predicts points near the ground-truth fracture surface boundary as fracture points. For the segmentation module, the precision remains while the recall may decrease a little bit. Nevertheless, our method effectively utilizes the primal-dual descriptor to establish matches between fracture points, as long as the local geometric primal feature can identify a unique correspondence in the dual feature space. In Figure 3, columns 2 and 3 visually depict that our method can successfully reconstruct the underlying object even when the fracture surface approximates a smooth curve.
> > >
> > > For the issue of comparison with registration baselines, in our comment to reviewer jmgU, we added preliminary results from SOTA registration baselines as the training was extremely time consuming. As the entire evaluation process progresses, we intend to provide more comprehensive and precise comparison results in this manuscript. The current results shows that registration baselines only yield poses that are close to random guesses, and are not applicable for the specific task of fracture assembly.
> > >
> > > We hope these comments adequately address your concerns.

---

### Official Review · Reviewer_LoVE · 2023-07-09

**Soundness:** 3 good
**Presentation:** 3 good
**Contribution:** 3 good
**Rating:** 6
**Confidence:** 4

**Summary:**

This work provides a Joint Learning of Segmentation and Alignment Framework(Jigsaw) framework for 3D fracture assembly task. Experiments are evaluted on Breaking Bad dataset and show better performance than other existing methods.

**Strengths:**

1. The proposed approach applies a joint learning framework specific for multi-part fracture assembly, which appends an attention-based
network module to capture local geometric information. To further capture viewpoint-dependent features for surface matching, the proposed method applies the primal-dual descriptor to achieve better matching.

2. The authors incorporate fracture point segmentation to capture intrinsic features and introduce a multi-part matching approach
for internal automatic piece positioning, with global pose alignment capability.

3. The proposed method shows better results than baseline methods evaluated on the multi-part assembly Breaking Bad dataset. The ablation analysis also validates the efficacy of the proposed network structure.

**Weaknesses:**

1.  The motivation of using primal-dual descriptor seems not sufficient, it would be better to append a more description of it.
2.  The ablation study part is considered a bit insufficient. e.g. The authors already mention matching loss and rigidity loss. The related ablation analysis about the impact of segmentation loss, matching loss and rigidity loss is missing.

**Questions:**

The running time analysis of the proposed method or the comparison to other methods is missing. How about this metric?

**Limitations:**

In the experiments, evaluation on extra dataset rather than the Breaking Bad dataset is much desired.

---

> ### Author Rebuttal · Authors · 2023-08-09
>
> Thanks for your time and valuable comments. Below we respond to your comments.
>
> > The motivation of using primal-dual descriptor
>
> **A**: The motivation is to capture the essence of the complementary geometry of two broken pieces. We provide two examples (Fig. 2 in the paper, Fig. 1 of the authors’ response document) where a conventional feature extractor falls short while the primal-dual descriptor excels. Please refer to the general response (II) for a more detailed explanation. We will add a more detailed explanation of the motivation of using primal-dual descriptors to the manuscript.
>
> > The ablation study part is considered a bit insufficient. e.g. The authors already mention matching loss and rigidity loss. The related ablation analysis about the impact of segmentation loss, matching loss and rigidity loss is missing.
>
> **A**: Both the segmentation loss and matching loss play essential roles in ensuring the assembly task's success. Omitting either of these components would render the method ineffective. Therefore, we compared the outcomes of applying these two loss functions jointly versus independently. We additionally investigate the integration of the rigidity loss. The results reveal a enhancement of $0.2 (\times 10^{-2})$ in the RMSE(T) metric.
>
> > The running time analysis of the proposed method or the comparison to other methods is missing.
>
> **A**: Please refer to the general response (I) for the running time of all the methods.

---

### Author Rebuttal · Authors · 2023-08-09

Dear area chair and reviewers,

We thank all the reviewers for the valuable comments. We appreciate that reviewers acknowledge that this paper is well-written, well motivated, and experimentally solid. Our proposed joint learning pipeline with the attention-based feature extractor, the fracture point segmentation, the primal-dual descriptor and multi-part matching reaches state-of-the-art performance for the multi-fracture assembly task, and are all recognized by the reviewers.

We begin with addressing some questions shared by reviewers.

**(I) The running time of Jigsaw compared to baseline methods:**

| Model     | Training Time / GPUs   | Speed (s/batch) / batch size       |
|-----------|:------------------------:|:----------------------------:|
| Global    | 21H           / 1 V100 |  1.65    /   32          |
| LSTM      | 16H           / 1 V100 | 1.63   / 32              |
| DGL       | 11H           / 1 V100 | 2.46    / 32             |
| Predator  | 96H           / 4 V100 | 1.28    / 28             |
| Jigsaw    | 120H          / 4 V100 | 7.67 / 8 * |

\* The time used in the forward is 1.3s/batch, the rest of the time is used on Hungarian and global alignment.


**(II) The motivation of primal-dual descriptor:**

When one object has been cracked into two pieces by physical force, the two pieces should exhibit complementary geometry. The conventional feature representation is used to find similar geometry and therefore would match two identical surfaces, which is undesirable in the context of fracture assembly. Our proposed primal-dual descriptor, however, is used to capture the essence of complementary geometry. It helps to foster correspondences between two parts that possess complementary attributes.

One example (Fig. 2 in the paper) is that a convex shape would be matched to another convex shape if only the conventional approach is used. Instead, our primal-dual descriptor could match it to a concave shape.
Another example (Fig. 1 of the authors' response document) showcases a situation that point $A$ would be matched to point $b$, and point $B$ would be matched to point $a$ with conventional approach. It causes two pieces to completely overlap. Though in the context of fracture assembly, with the proposed primal-dual descriptor, point $A$ could be correctly matched to point a and point $B$ to point $b$.

The learning of two descriptors should be viewed in conjunction with the construction of the affinity metric (Eq. (6)) and the learning procedure guided by the matching loss (Eq. (7)). By providing ground truth matching for complementary points, the system would favor the similarity between complementary attributes over the identical geometries. As demonstrated in Fig. 6 of the appendix, our primal-dual descriptor refrains from self-matching and finds matching across disparate fragments. It is an experimental evidence showing the primal-dual system can capture distinct features for one fracture point.

For other issues that require clarification, we provide detailed answers in the individual point-by-point response.

---

### Decision · Program_Chairs · 2023-09-21

**Decision:**

Accept (poster)

**Comment:**

This paper introduces intriguing work for assembling fractured parts. Reviewers favor the proposed approach and the demonstrated results. The authors provided comprehensive feedback regarding specific questions, and the authors provided a supplement pdf file. In particular, there were concerns from reviewer jmgU on the practical aspects, but AC notes that the authors adequately defend it. AC agrees that assembling fractured parts has very small overlaps between parts, and it is distinct from pair-wise point cloud registration problems. AC also notes that the proposed primal-dual descriptor is an interesting attempt. After the rebuttal phase, most of the reviewers voted for the positive scores. As a result, AC recommends paper acceptance. It is strongly advised to apply the valuable discussions and additional materials in the revision.